# Knowledge Distillation as Decontamination? Revisiting the "Data Laundering" Concern in Classification Tasks

**Hengyu Luo[1], Raúl Vázquez[1], Timothee Mickus[1], Filip Ginter[2,3], Jörg Tiedemann[1,3]**
[1]University of Helsinki, [2]University of Turku, [3]ELLIS Institute Finland
[1]`firstname.lastname@helsinki.fi`  [2]`figint@utu.fi`

## Abstract

Concerns have been raised that knowledge distillation may transfer test-set knowledge from a contaminated teacher to a clean student—a "data laundering" effect that potentially threatens evaluation integrity. In this paper, we assess the severity of this phenomenon. If these concerns regarding data laundering are minor, then distillation could be used to mitigate risks of direct data exposure. Across eight classification benchmarks, we find that substantial laundering is the exception rather than the rule: unlike the large performance gains from direct contamination, any accuracy inflation from laundering is consistently smaller and statistically insignificant in all but two cases. More broadly, using sample-level analysis, we find that the two phenomena are weakly correlated, suggesting that laundering is not simply a diluted form of contamination but a distinct effect that arises primarily when benchmarks exhibit large train–test distribution gaps. Motivated by this, we conduct controlled experiments that systematically enlarge the train–test distance on two benchmarks where laundering was initially negligible, and observe that laundering becomes more significant as the gap widens. Taken together, our results indicate that knowledge distillation, despite rare benchmark-specific residues, can be expected to function as an effective decontamination technique that largely mitigates test-data leakage. Code is available at `https://github.com/hengyu-luo/kd-revisiting-data-laundering-concern`.

## 1 Introduction

Proprietary models have been proven to, perhaps inadvertently, learn from leaked benchmark data, raising questions about the reliability of closed-source models (Magar & Schwartz, 2022; Balloccu et al., 2024). One particularly subtle form of contamination is **data laundering**. Distinguished from **direct contamination, which we define as the scenario where a model creates a training-time shortcut by memorizing exact test samples present in its training corpus,** data laundering is where test-set knowledge leaks to a student model during knowledge distillation via a contaminated teacher, compromising evaluation integrity (Mansurov et al., 2025). While prior work highlighted this phenomenon, the prevalence, magnitude, and mechanisms of laundering remain largely unexplored. To assess the critical risk of this phenomenon, we ask: is data laundering a pervasive threat that undermines current benchmarking practices? We focus on classification tasks due to two factors: first, it offers a relatively straightforward setup that is likely to provide a reasonable approximation of the effects of laundering in a broader scope of settings, from sequence generation to ranking; second, many modern applications of NLP technology (from NER to WSD) are still explicitly framed as classification problems. Our extensive experiments on classification tasks suggest that data laundering is often much weaker than direct contamination and can even mitigate some of its harmful effects. This work contributes a potential foundation for establishing **safer empirical research environments**.

First, we conduct a large-scale assessment across eight benchmarks to determine the prevalence and magnitude of data laundering. We find that significant laundering is a rare phenomenon: while it does occur, its effect on model accuracy is substantially smaller than that of direct contamination, and in many cases the difference is not statistically significant. This initial finding suggests that knowledge distillation may indeed function as an effective decontamination method.

Given that data laundering effects are substantially weaker than those of direct contamination, we investigate whether it is simply a watered-down form of direct contamination or a distinct phenomenon. Using sample-level analysis across the same benchmarks, we examine whether a sample's sensitivity to direct contamination predicts its sensitivity to laundering. Our analysis reveals only a weak correlation between the two; samples highly susceptible to direct contamination are not necessarily the ones most affected by laundering. This suggests that laundering is indeed a distinct phenomenon.

Finally, we explore the conditions under which data laundering emerges. Controlled experiments show that systematically widening the train-test distributional gap increases the effects of laundering, suggesting a causal connection. Consistent with these experiments, benchmarks with naturally larger train-test gaps tend to exhibit stronger laundering effects, although the magnitude and statistical significance of the effect vary across datasets. Our findings indicate that both the characteristics of the benchmark and the degree of train-test shift influence the extent of data laundering, highlighting that its occurrence depends on benchmark-specific factors such as dataset domain and the degree of train–test distributional shift, rather than being a universal consequence of knowledge distillation.

In summary, this paper systematically disentangles the role of data laundering in model evaluation. We argue that while the concern is valid, its practical impact is often minimal. Our findings indicate that laundering is generally limited in scope, substantially smaller than direct contamination, and tied to particular conditions such as train–test distributional shifts. By contextualizing these risks and identifying the conditions under which they arise, we provide a pathway for more responsible and reliable model evaluation including a principled use of KD in the era of ubiquitous large models.

## 2  RELATED WORK

**Data contamination in evaluation.** A growing number of work has shown that benchmark integrity can be compromised when test material leaks into pretraining or fine-tuning corpora, artificially inflating scores without corresponding generalization. Early red flags already appeared with large web-scale LMs such as GPT-3 (Brown et al., 2020) and in corpora audits like C4 (Dodge et al., 2021), while Magar & Schwartz (2022) provided a controlled, task-level analysis linking memorization to performance inflation. Subsequent studies proposed black-box and white-box detectors—e.g., guided-instruction "time travel" tests (Golchin & Surdeanu, 2024) and distributional peakedness checks (Dong et al., 2024)—and documented practical challenges for closed models (OpenAI, 2023). Broader surveys and empirical audits emphasize that overlap can be subtle (paraphrases, partial spans, synthetic rephrasings) and uneven across benchmarks, motivating routine, benchmark-specific contamination checks (Sainz et al., 2023) and calls for provenance transparency (e.g., to report train–test overlap; Zhang et al., 2025). Recent work also targets modern LLM benchmarks directly, offering methods tailored to both open and proprietary models (Deng et al., 2024).

**Knowledge distillation and data laundering.** Knowledge distillation (KD) is a standard tool for compression and transfer (Hinton et al., 2015; Sanh et al., 2019), but it also opens a distinct vector for leakage. Mansurov et al. (2025) formally introduced *data laundering* showing that a contaminated teacher (exposed to test data) can pass benchmark-specific knowledge to a student trained only on clean data via KD, inflating evaluation without direct access to the test set. However, their study had limitations: the experiments relied on a bert-base-uncased student trimmed down to just 2 layers, rather than using a pretrained 2-layer model, making results difficult to disentangle from near-random baselines. Additionally, the study did not compare laundering against direct contamination or systematically explore when it arises. As a result, the prevalence, magnitude, and mechanisms of laundering remain unclear, motivating our more systematic analyses. Complementary evidence from ranking distillation shows that even tiny teacher exposure (e.g., $< 0.1\%$ of training) can yield inflated student effectiveness, especially with pairwise/order-based objectives (Suresh Kalal et al., 2024). Security-oriented studies further show that KD can transmit non-benign artifacts (e.g., backdoor behaviors), particularly in data-free settings (Hong et al., 2023). Together these results underscore procedural defenses such as transparent training histories and contamination-aware KD protocols (Zhang et al., 2025).

## 3 METHODOLOGY

### 3.1 EXPERIMENTAL SETUP AND DATA

**Models and Distillation Process**    To isolate and measure data laundering, we use a controlled, two-stage process. Our setup involves fine-tuning eight models in total for each benchmark. First, we train teacher models using `bert-base-uncased` (Devlin et al., 2019). A **clean teacher** ($T_{clean}$) is fine-tuned on the original training set, while a **dirty teacher** ($T_{dirty}$) is fine-tuned on a training set contaminated with test data. Besides, to validate the generality of our findings across different architectures, we also include modern large language models, namely `Llama3.2-1B` (Grattafiori et al., 2024) and `Qwen3-0.6B` (Yang et al., 2025), as teacher models. These models represent the current state-of-the-art in lightweight LLMs.

Second, we use these teachers to distill knowledge into smaller student models (`distilbert-base-uncased`) (Sanh et al., 2019) using soft-label distillation with forward KL divergence (Wang et al., 2024). Crucially, the distillation process for both student types ($S_{clean}$ distilled from $T_{clean}$ and $S_{dirty}$ from $T_{dirty}$) is always performed using the original, clean training set. This ensures that any test-set knowledge is transferred exclusively via the teacher model, not through direct data exposure during the student's training.

To serve as a control group and contextualize the results, we also establish baseline models that share the same architecture as the students. A **Clean Baseline** ($B_{clean}$) and a **Dirty Baseline** ($B_{dirty}$) are created by fine-tuning the student architecture directly on the clean and contaminated datasets, respectively.

**Benchmarks and Contamination Protocol**    We selected eight public classification benchmarks, with diverse domains ranging from topic classification, sentiment analysis, and intent recognition to emotion detection, and NLI (details in Appendix B.2, Table 5). This diversity ensures that our findings generalize across tasks with varying difficulty, number of labels, and dataset sizes. For each benchmark, we create a contaminated dataset to train $B_{dirty}$ and $T_{dirty}$ with a **replacement strategy**: we contaminate training data by injecting the full test set and removing an equal number of original training samples to keep size constant. All experiments are repeated with five different random seeds to ensure robustness.

### 3.2 EVALUATION METRICS

We employ a set of metrics to quantify data leakage, distinguishing between those that operate on the entire benchmark and those that apply to individual samples. Let the test set be denoted by $\mathcal{C} = \{x_1, x_2, \ldots, x_n\}$. Metrics computed over $\mathcal{C}$ capture the overall impact of data leakage on model accuracy at the benchmark level, and sample-level metrics evaluate performance on individual examples $x_i$.

**Sample-Level Leakage Scores**    To analyze leakage mechanisms at a finer granularity, we measure how much each test sample $x_i$ becomes easier or harder under different training conditions. The *difficulty* of a sample $x_i$ under a model $M$ is defined as

$$D(x_i, M) = 1 - P(y_i \mid x_i; M),$$

that is, the probability of the model assigning the wrong label. Intuitively, higher $D(x_i, M)$ means the model finds $x_i$ more difficult.

We then define two sample-level leakage effects:

$$\Delta_{\text{laund}}(x_i) = D(x_i, S_{\text{dirty}}) - D(x_i, S_{\text{clean}}),$$
$$\Delta_{\text{contam}}(x_i) = D(x_i, B_{\text{dirty}}) - D(x_i, B_{\text{clean}}).$$

Here $\Delta_{\text{laund}}$ captures how much a student model changes when trained on dirty vs. clean teachers, while $\Delta_{\text{contam}}$ captures how much a baseline model changes when directly trained on dirty vs. clean data. These sample-level scores provide a natural way to capture the effect of laundering and contamination on individual examples. In practice, we typically expect these $\Delta$ values to be negative, since the presence of laundering or contamination generally reduces the sample difficulty.

**Laundering vs. Contamination Correlation** To assess whether data laundering and direct contamination are mechanistically related, we compute a benchmark-level correlation from the sample-level scores. For a given test set $\mathcal{C}$, we first construct two vectors of leakage effects:

$$\mathbf{l} = [\Delta_{\text{laund}}(x_1), \dots, \Delta_{\text{laund}}(x_n)], \quad \mathbf{c} = [\Delta_{\text{contam}}(x_1), \dots, \Delta_{\text{contam}}(x_n)].$$

We then calculate the Pearson correlation coefficient

$$r(\mathcal{C}) = \frac{\text{cov}(\mathbf{l}, \mathbf{c})}{\sigma_{\mathbf{l}} \, \sigma_{\mathbf{c}}}. \tag{1}$$

This metric is advantageous as it is scale-invariant, allowing us to compare the directional agreement of the two phenomena across all samples, irrespective of the absolute magnitude of their effects.

## 4 RESULTS AND ANALYSIS

### 4.1 KNOWLEDGE DISTILLATION AS AN EFFECTIVE DECONTAMINATION TECHNIQUE

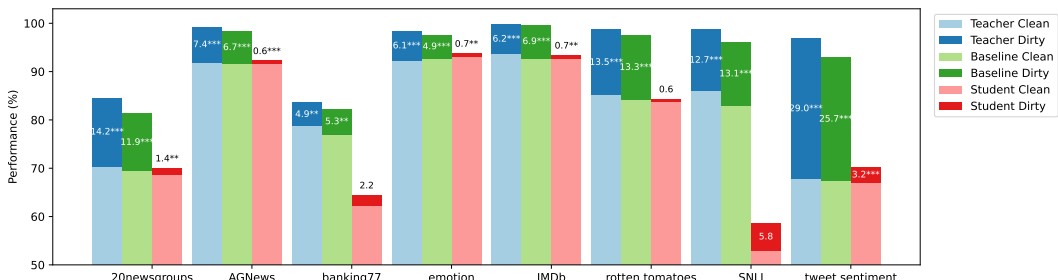

Figure 1: Performance on clean and contaminated benchmarks for Teacher, Baseline, and Student models. Bars show clean (lighter) and contaminated/dirty (darker) accuracies, with $\Delta\text{Acc}_M$ values annotated. The statistical significance of the differences are marked with $^*$p<0.05, $^{**}$p<0.005, $^{***}$p<0.001.

We establish an upper bound on potential data leakage by comparing the aggregate impact of contamination using benchmark-level metrics. Specifically, we contrast the gains from direct contamination in clean and dirty baseline ($B$) and teacher ($T$) models with the gains from data laundering in clean and dirty student ($S$) models. For a model $M$ and benchmark $\mathcal{C}$, the performance gain due to test-set leakage at training time is defined as

$$\Delta\text{Acc}_M = \text{Acc}(M_{\text{dirty}}, \mathcal{C}) - \text{Acc}(M_{\text{clean}}, \mathcal{C}). \tag{2}$$

When $M = B$ or $M = T$, Equation (2) measures the direct contamination effects, whereas for $M = S$ it measures data laundering gains.

Figure 1 presents the performance gains over the baselines for all models (the full results are reported in Appendix C, Table 8). Comparing the gains of baseline and student models allows us to quantify the relative impact of data laundering on observed performance improvements, and contrasting the direct contamination gains of baseline and teacher models highlights the impact of model capacity.[1]

The results reveal a clear and consistent pattern: directly training on a test-set-contaminated dataset results in substantial and highly significant performance gains across all eight benchmarks for all Teacher and Baseline models. Performance gains for the Baseline models range from $4.89\%$ on `emotion` to $25.66\%$ on `tweet_sentiment`, confirming the well-documented effects of direct data contamination and providing a crucial reference point for evaluating the impact of knowledge distillation.

---

[1]An exception is `SNLI`, where the performance gap between the clean student and the clean baseline is unusually large. This result may be explained by two factors: (i) all models were trained for only 3 epochs without early stopping to ensure fair comparison with the teacher and baseline, which may have left the student undertrained when relying solely on teacher signals; and (ii) `SNLI` is inherently more challenging than typical classification tasks, making it harder for the student to achieve strong performance under distillation.

By contrast, the gains observed due to data laundering ($\Delta \text{Acc}_S$) are noticeably smaller, highlighting the mediating effect of knowledge distillation. For example, on `tweet sentiment`, the Baseline's $25.66\%$ gain is reduced to $3.25\%$ after distillation through a contaminated Teacher. Similarly, on `20newsgroups`, an $11.91\%$ direct gain shrinks to just $1.42\%$. This trend is consistent across all benchmarks: distillation acts as a strong bottleneck, significantly mitigating the performance inflation caused by direct contamination. Overall, these results suggest that knowledge distillation, rather than solely propagating leakage, may act as a *effective decontamination mechanism*, substantially mitigating the performance inflation caused by direct contamination.

The gains from data laundering are clearly more modest than those from direct contamination, with significant increases on datasets like `agnews` ($0.65\%$) and `tweet_sentiment` ($3.25\%$). However, for three benchmarks (`banking77`, `rotten tomatoes`, and `SNLI`) the difference in performance is not statistically significant. Furthermore, to assess the practical significance of these gains, we also computed bootstrap confidence intervals (see Appendix B.4), which provided further confirmation of the observed effects.

This detailed observation allows us to refine our perspective. While the concern about data laundering is not unfounded, as the phenomenon does occur, its practical impact is the exception rather than the rule. It is a rare and mild effect, with a smaller magnitude and limited influence on overall model evaluation.

**Robustness across Distillation Strategies**    To examine the robustness of our findings, we further extended our analysis across diverse distillation objectives, including Soft Forward, Soft Reverse, and Hard-label distillation under both pure and mixed loss formulations. We observe modest variations in vulnerability: mixed objectives and soft labels can slightly increase the impact of data laundering. However, the overall pattern remains consistent: knowledge distillation acts as an effective decontamination filter in classification tasks. Across all investigated strategies, statistically significant data laundering remains an exceptional and low-magnitude phenomenon compared to direct contamination. We provide a detailed breakdown of these results and the associated trade-offs in Appendix C.2.

**Generality across Modern Architectures**    We further validated our findings using Llama-3.2-1B and Qwen3-0.6B as teachers, selecting four representative tasks from our benchmark suite: AGNews, Emotion, Rotten Tomatoes, and Tweet Sentiment. In these experiments, the student model remains DistilBERT trained with Soft Forward distillation ($\alpha = 1$, $\tau = 2.0$), ensuring that any differences reflect the teacher's influence rather than changes to the student architecture. Detailed experimental setups and the full set of results can be found in Appendix C.3 (Table 12).

Figure 2 presents the performance of the student under Llama and Qwen teachers, showing both the clean and contaminated teacher models, along with the corresponding student outcomes. While the LLM teachers achieve near-perfect accuracy on some contaminated benchmarks (e.g., Tweet Sentiment and Rotten Tomatoes), student gains from direct contamination remain minimal, consistent with the observations from BERT-based teachers. For instance, while the Llama-3.2 teacher gains $25.80\%$ on Tweet Sentiment, the student gain is limited to $1.91\%$, underscoring the informational bottleneck imposed by distillation.

These results highlight that switching to a larger, more capable teacher has minimal effect on student contamination, reinforcing our conclusion that KD effectively limits the propagation of memorized data. However, we emphasize that these preliminary experiments are to be interpreted with caution. It is likely that the test sets overlap with the LLMs' pretraining data, which inflates observed gains and undermines our attempts to keep our experiments under a controlled scenario.

Nevertheless, as shown in Table 1 the choice of teacher produces relatively small differences in student models under our "clean student" setup. Even if we treat Llama or Qwen as effectively dirty teachers (given their likely exposure to our benchmarks), comparing them to our guaranteed clean teacher constitutes a clean-vs-dirty scenario. Under this interpretation, the lack of substantial gains in students distilled from LLM teachers is consistent with our earlier conclusions. Data laundering has a limited impact and KD can serve as a buffer, even in the presence of strong teachers.

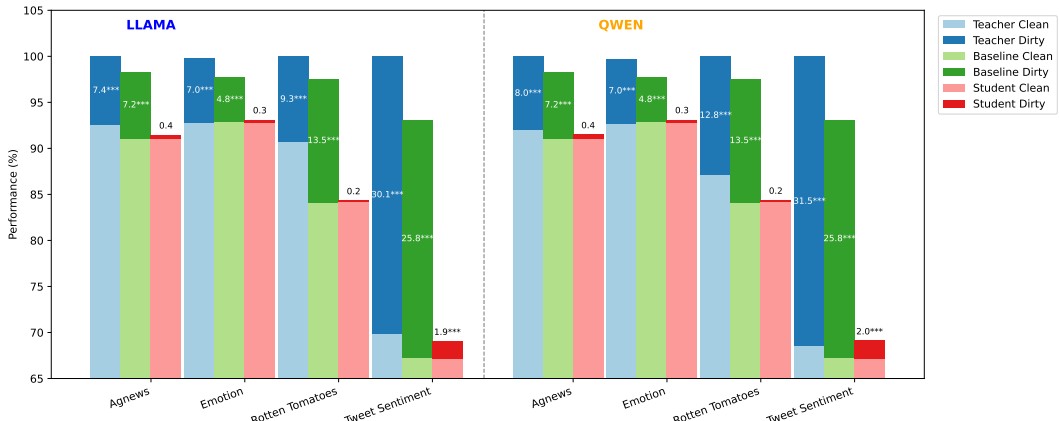

Figure 2: Performance on clean and contaminated benchmarks for LLM Teachers, Baseline, and Student models. The teachers used are Lama-3.2-1B and Qweb3-0.6B. Bars show clean (lighter) and contaminated/dirty (darker) accuracies, with $\Delta\text{Acc}_M$ values annotated. The statistical significance of the differences are marked with $^*$p<0.05, $^{**}$p<0.005, $^{***}$p<0.001.

| Teacher Model | agnews | emotion | rotten tomatoes | tweet sentiment |
|---|---|---|---|---|
| BERT-base-uncased | 91.59 | 93.10 | 83.77 | 66.99 |
| LLAMA3.2-1B | 91.08 (-0.51) | 92.81 (-0.29) | 84.15 (+0.38) | 67.13 (+0.14) |
| QWEN3-0.6B | 91.49 (-0.10) | 92.82 (-0.28) | 84.22 (+0.45) | 67.10 (+0.11) |

Table 1: Performance comparison of the clean students $\left(S_{clean}^{soft,fwd}\right)$ when trained with different teachers. The numbers in parentheses indicate the gain (or loss) relative to the BERT student.

## 4.2 LAUNDERING AND CONTAMINATION: A TALE OF TWO MECHANISMS

Having established that data laundering is a rare and much weaker effect than direct contamination across diverse architectures and objectives, we now turn to dissecting the mechanism itself. To conduct a granular, sample-level analysis in a strictly controlled environment, we return to our primary experimental configuration: a BERT-based teacher with Soft Forward distillation. Under this canonical setup, a crucial question arises regarding the nature of the phenomenon: is laundering merely a weaker, scaled-down version of direct contamination, or is it a distinct phenomenon with its own underlying mechanism? If it were simply "contamination-lite," we would expect the samples most affected by both phenomena to be highly correlated.

| $\mathcal{C}$ | 20newsgroups | agnews | banking77 | emotion | imdb | rotten tomatoes | snli | tweet sentiment |
|---|---|---|---|---|---|---|---|---|
| $r(\mathcal{C})$ | 0.30(03)*** | 0.32(02)*** | 0.13(08) | 0.26(12)*** | 0.30(02)*** | 0.12(06)* | -0.03(17)*** | 0.31(01)*** |

Table 2: Benchmark correlations between laundering and contamination effects. Shows baseline accuracy ($B_{\text{clean}}$) and Pearson correlations of sample-level effects. Statistical significance was assessed using bootstrapping-based tests, with detailed procedures provided in Appendix B.4.

In Table 2, we show the Pearson correlation between the sample-level laundering effect and contamination effect scores. We find that on the benchmarks most susceptible to laundering, agnews and tweet_sentiment, the correlations are as small as 0.32 and 0.31, respectively. While statistically significant, these values are far below the commonly accepted threshold of 0.7 for a strong relationship (see e.g., Rickert et al., 2023; Kjell et al., 2022), indicating only a weak linear association. The connection is weaker still on other benchmarks, and even becomes slightly negative for snli.

This weak correlation suggests that the two phenomena impact samples differently. A sample that is highly vulnerable to being "memorized" through direct training is not necessarily the same sample whose knowledge is indirectly transferred through a teacher model. This observation motivates a deeper exploration into the nature of these mechanisms. To explore this further, we conduct a granular, sample-level analysis on the tweet_sentiment benchmark, where the laundering effect was most pronounced. Figure 3 visualizes the laundering and contamination effects for each test sample, sorted by their difficulty as perceived by both the clean baseline and clean student models.

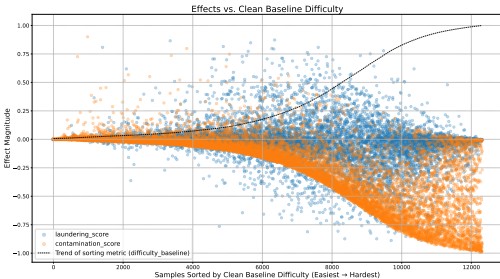 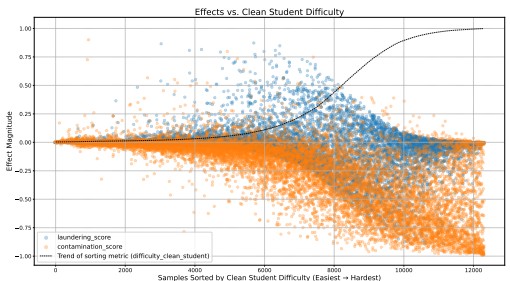

(a) Samples sorted by clean baseline difficulty     (b) Samples sorted by clean student difficulty

Figure 3: Laundering and contamination effects on the `tweet_sentiment` benchmark, with samples sorted by difficulty. The contamination effect (orange) shows a strong, monotonic downward trend as samples become harder. In contrast, a laundering effect (blue) is more dispersed, has a much weaker trend, and is less correlated with initial sample difficulty.

By sorting samples by the difficulty perceived by the clean model, we can visualize that some samples are more inherently prone to contamination than others: those that the clean model finds more difficult have a larger potential for improvement when seen during training. As a result, the contamination effect (in orange) exhibits a strong and relatively monotonic downward trend—a more negative $\Delta_{\text{contam}}$—on harder samples. This is an expected signature of direct test-set exposure. In stark contrast, the data laundering effect (in blue) does not share this strong monotonic relationship with sample difficulty. Its overall magnitude is smaller, its trend is weaker, and it exhibits significant volatility, with benefits appearing for both difficult and some easy samples.

Together, the weak sample-level correlations and non-monotonic laundering trends show laundering is not "scaled-down contamination" but a distinct mechanism. Instead, it is an independent mechanism triggered by different conditions. This discovery leads to the conclusion that data laundering, when it occurs, is a more elusive mechanism than direct contamination and likely possesses its own unique set of enabling conditions.

## 5 DISCUSSION: THE ROLE OF DISTRIBUTIONAL GAPS

Our analysis in the previous section established that data laundering is a rare, mild, and mechanically distinct phenomenon from direct contamination. This prompts the next logical question: what specific characteristics of a benchmark make it more susceptible to this elusive effect? Our results revealed the impact of data laundering to be highly benchmark-specific, with datasets such as `tweet_sentiment` exhibiting comparatively more significant effects. This naturally leads us to hypothesize that some intrinsic property of these datasets may be at play.

| Dataset | Jaccard | | TF-IDF | | Avg Emb Sim | | Average Max Semantic Sim | | Average Pattern Conformity | |
|---|---|---|---|---|---|---|---|---|---|---|
| | Micro | Macro | Micro | Macro | Micro | Macro | Micro | Macro | Micro | Macro |
| agnews | 0.1686 | 0.1463 | 0.9881 | 0.9661 | 0.9984 | 0.9965 | 0.6682 | 0.6554 | 0.4942 | 0.5468 |
| **tweet_sentiment** | **0.0887** | **0.0733** | **0.6310** | **0.6371** | **0.7115** | **0.7681** | **0.5102** | **0.4913** | **0.4437** | **0.4523** |
| 20newsgroups | 0.1787 | 0.1513 | 0.9848 | 0.8559 | 0.9921 | 0.9599 | 0.5985 | 0.5709 | 0.4871 | 0.5431 |
| emotion | 0.0941 | 0.0666 | 0.9820 | 0.9072 | 0.9984 | 0.9877 | 0.6403 | 0.5954 | 0.5504 | 0.5622 |
| imdb | 0.2218 | 0.2116 | 0.9978 | 0.9961 | 0.9994 | 0.9990 | 0.6919 | 0.6837 | 0.6669 | 0.6707 |
| rotten_tomatoes | 0.0702 | 0.0606 | 0.9198 | 0.8609 | 0.9987 | 0.9975 | 0.6331 | 0.6229 | 0.5920 | 0.5958 |
| snli | 0.1359 | 0.1552 | 0.9906 | 0.9877 | 0.9979 | 0.9975 | 0.7011 | 0.6668 | 0.5606 | 0.5605 |
| banking77 | 0.2874 | 0.2217 | 0.9810 | 0.9103 | 0.9963 | 0.9887 | 0.8985 | 0.8940 | 0.7520 | 0.8627 |

Table 3: Similarity metrics between the test set and the training subsets. "Micro" refers to the global similarity, calculated across all samples without referencing labels. "Macro" refers to the unweighted average of similarities computed on a per-label basis. Benchmarks more vulnerable to data laundering, such as `tweet_sentiment`, happen to appear lower similarity scores.

To investigate this, we first characterize the intrinsic relationship between the training and test sets in a model-agnostic way. We compute a suite of similarity metrics, all normalized to the range $[0, 1]$, to help identify inherent data properties that might make a benchmark susceptible to leakage. The metrics used are: Jaccard Similarity, TF-IDF Cosine Similarity, Average Embedding Similarity,

Average Max Semantic Similarity, and Average Pattern Conformity. The detailed mathematical formulations for these metrics are provided in Appendix B.5.

An analysis of these metrics across the benchmarks (detailed in Table 3) revealed a notable pattern. The benchmarks most affected by laundering (e.g., `tweet_sentiment`) consistently exhibit lower similarity scores across several metrics. This indicates a larger distributional gap between their training and testing sets. This observation provides us with a plausible hypothesis: data laundering is more likely to occur, and its effects are more pronounced, when there is a significant distributional distance between a benchmark's training and test sets.

The intuition behind this is that when a teacher model is contaminated with test data that deviates from the training data's dominant semantic patterns, it is exposed to alternative, test-specific regularities. These regularities different from those emphasized in the training distribution can then be systematically learned by the teacher. Crucially, such patterns may be particularly advantageous for the test set, and can subsequently be passed on to the student during distillation. Based on this hypothesis, we designed a series of controlled experiments to systematically validate this relationship.

## 5.1 EXPERIMENTAL DESIGN

To test our hypothesis rigorously, we designed a controlled experimental setup to systematically vary the train-test distribution gap. This section outlines the creation of our stratified datasets and the experimental protocol.

**Creating Controlled Distributional Gaps** The core of our methodology involves partitioning the training data of the `emotion` and `rotten_tomatoes` benchmarks into distinct subsets. This is achieved through a **stratified splitting** process: for each class, we first identify its test-set centroid, and then partition the training samples belonging to that class into five equal-sized quintiles based on their semantic similarity to this centroid. These quintiles are then aggregated across all classes to yield five global training sets, denoted as Levels 1 through 5. **Level 1** contains samples most similar to the test set (smallest gap), while **Level 5** contains those least similar (largest gap). This method successfully creates the intended gradient of distributional gaps. The effectiveness of this partitioning is validated by a systematic decrease in cosine similarity from Level 1 to 5, as well as a consistent monotonic decrease across five other similarity metrics (see Appendix B.6 for detailed visualizations).

**Training and Contamination Protocol** For each of the five data levels, we conduct a consistent training and distillation protocol. We train a **clean teacher** ($T_{\text{clean}}$) on the original training quintile and a **dirty teacher** ($T_{\text{dirty}}$) on its contaminated counterpart. We use an `add` mode (as opposed to `replace` mode) for contamination in this setup; because each training level is significantly smaller than the original dataset, this approach ensures the model has sufficient data for robust training and mitigates the impact of the reduced training set size. Knowledge from both teachers is then distilled into respective student models, $S_{clean}$ and $S_{dirty}$. The distillation process itself always uses the clean training data of that level, isolating the transferred knowledge as the primary variable.

## 5.2 RESULTS: LAUNDERING EFFECT INTENSIFIES WITH WIDER DISTRIBUTIONAL GAPS

With the controlled gaps established and verified, we now turn to the central result of our experiment. As shown in Figure 4, our findings demonstrate a clear relationship between the train-test distributional gap and the manifestation of data laundering. Specifically, we observe that as the distributional gap widens, the data laundering effect becomes **statistically more significant and pronounced**, and in some cases, increases in magnitude.

**Direct Contamination Effect Remains Stable** First, we observe that for both the `emotion` and `rotten_tomatoes` datasets, a significant direct contamination effect is present from Level 1 to 5, evidenced by the performance differences between the dirty and clean teachers, as well as between the dirty and clean baselines. However, this performance gap does not show significant fluctuation or a clear trend as the distributional gap widens. This can be observed visually in Figure 4, where the gaps between clean and dirty models (for both teachers and baselines) remain largely stable across the

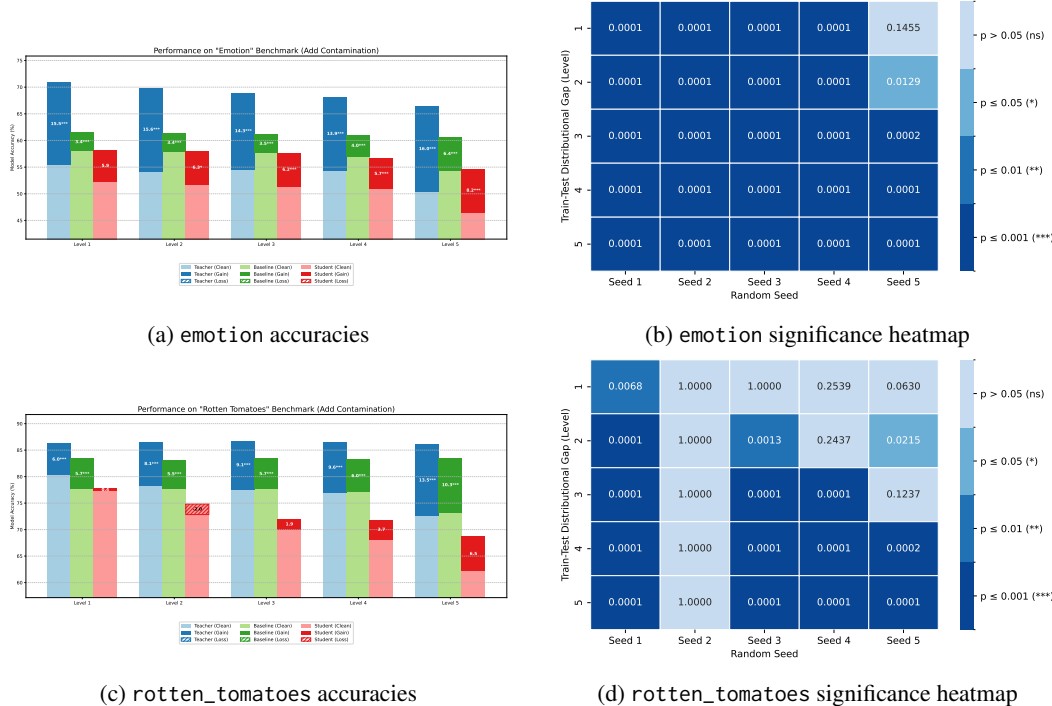

(a) emotion accuracies

(b) emotion significance heatmap

(c) rotten_tomatoes accuracies

(d) rotten_tomatoes significance heatmap

Figure 4: Data laundering effects across controlled distributional gaps for emotion and rotten_tomatoes. Left panels: Model accuracy comparison, where the performance gain for the student model (red bars, $\Delta\text{Acc}_S$) represents the laundering effect. Right panels: Heatmaps of p-values testing the significance of the laundering gain over 5 random seeds. The increasing darkness from Level 1 to 5 indicates that the **statistical significance of the laundering effect becomes more pronounced** as the train-test distributional gap widens, even if the absolute magnitude of the gain varies. Statistical significance ($*p < 0.05, **p < 0.01, ***p < 0.001$) was assessed using bootstrapping-based tests.

levels. Our experiment thus indicates that, under this setup, the magnitude of the direct contamination effect is not significantly correlated with the distributional gap between the training and test sets.

**Analysis of the emotion Benchmark: Increased Statistical Certainty**   We first examine the emotion dataset. As seen in the bar charts of Figure 4(a), the absolute performance gap between clean and dirty students ($\Delta\text{Acc}_S$) remains relatively small across all levels. However, the significance heatmap in Figure 4(b) reveals a crucial trend: the p-values systematically decrease as we move from Level 1 (smallest gap) to Level 5 (largest gap). This indicates that while the magnitude of the leakage may not drastically increase, the statistical certainty of its existence strengthens. In other words, large distributional gaps transform laundering from a rare and inconsistent occurrence into a consistent, statistically significant phenomenon, even if the resulting accuracy gain is modest.

**Analysis of the rotten_tomatoes Benchmark: Growth in Both Significance and Magnitude** The rotten_tomatoes benchmark exhibits an even more pronounced pattern. Consistent with the trend seen in emotion, the statistical significance of the laundering effect increases as the distributional gap widens (Figure 4(d)), with p-values consistently dropping to highly significant levels ($p < 0.01$ or $p < 0.001$) at larger gaps. Moreover, unlike emotion, this benchmark also shows a visible widening of the performance delta between clean and dirty students as the level increases (Figure 4(c)). This suggests that for certain datasets, a larger distributional gap not only increases the likelihood of laundering occurring but can also increase its impact on model performance.

**Summary and Implications**   Our results identify the train-test distributional gap as a key factor driving data laundering, increasing its statistical consistency even where absolute magnitude varies.

In contrast, direct contamination shows no such correlation. This finding reveals a trade-off in benchmark design: the distributional distance required to evaluate generalization may inadvertently increase laundering risks. Consequently, we argue that the rigid single-test-set paradigm is insufficient. Instead, employing multiple test sets at varying distributional distances offers a more robust approach, simultaneously assessing generalization power and resilience to data laundering.

**Practical Guidelines for Benchmark Release**   We propose a rigorous two-step curation protocol based on these findings: (1) Diagnostic Gap Measurement: Prior to release, curators should quantify train–test shifts using the lexical and semantic metrics detailed in Appendix B.5. Consistently low similarity scores serve as a diagnostic flag for high laundering susceptibility. (2) Distance-Aware Evaluation: If a large gap is detected, we recommend releasing multiple test splits positioned at varying distributional distances. This enables a systematic audit of knowledge distillation systems to distinguish between true robust generalization and artifacts of data laundering driven by specific distributional shifts.

## 6   CONCLUSION

This paper set out to investigate the severity of **data laundering** and its implications for the integrity of classification evaluation. Our comprehensive investigation across eight benchmarks offers a reassuring, if nuanced, conclusion: concerns of data laundering as a pervasive threat appear largely overstated. Instead, we find that **knowledge distillation appears to be a promising decontamination technique**, dramatically attenuating the performance inflation caused by direct test-set exposure.

While distillation acts as a strong buffer, it is not a perfect one. We confirm that residual leakage can occur, but these instances of significant laundering are the exception, not the rule. Crucially, our analysis reveals that this leakage is not simply a diluted form of direct contamination but a **mechanically distinct phenomenon**. We identified the **train-test distributional gap** as a key driver, a hypothesis confirmed through controlled experiments where systematically widening this gap induced a significant laundering effect.

In short, knowledge distillation, far from being a liability, is a robust defense against test-data leakage in classification settings. The rare, benchmark-specific instances of laundering are not an indictment of the method, but a predictable consequence of large distributional shifts between training and test data—a factor that benchmark designers may need to consider. Our findings thus help clarify the risks associated with distillation and provide a path toward more reliable and responsible model evaluation.

## 7   LIMITATIONS AND FUTURE WORK

**Limitations**   Our work is primarily constrained by its focus on BERT-family encoder-only classification models, leaving the effects on larger-scale models and different student architectures, such as decoder-only LMs, unexplored. Besides, the analysis was confined to classification tasks; data laundering in generative contexts, where distillation and contamination evaluation strategy can be more diverse and complex, even in scenarios where teachers generate synthetic data (data-free KD), remains an open question. Finally, our experiments used only English datasets, so the findings may not generalize to multilingual or domain-specific scenarios.

**Future Work**   These limitations suggest new research avenues. Future work should extend our work to broader architectures, training regimes and tasks. A key step require extensive efforts will be the development of appropriate metrics for generative models: in the present state, our study does not addres text-generation systems, which is an import point to cover given the the current research landscape. It is also crucial to examine data laundering on multilingual and domain-specific benchmarks to test the generality of our findings and to further probe the mechanistic distinctions between laundering and direct contamination. While our current full test-set replacement constitutes a controlled worse-case scenario, studying contamination under training regimes that resemble how leakage occurs in the wild (typically, partial, noisy, or indirect) remains an important direction to study. Besides, a particularly promising direction involves leveraging KD for decontamination use, and even act as a diagnostic tool to infer the contamination level of teacher models.

ACKNOWLEDGMENTS

This project is funded by the AI-DOC program hosted by Finnish Center of Artificial Intelligence (decision number VN/3137/2024-OKM-6). This research is also supported by the OpenEuroLLM project, co-funded by the Digital Europe Programme under GA no. 101195233. The authors wish to acknowledge CSC - IT Center for Science, Finland, and LUMI supercomputers, owned by the EuroHPC Joint Undertaking, for providing computational resources.

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

## A  STATEMENTS

### A.1  ETHICS STATEMENT

The datasets used in this work do not involve any sensitive or personally identifiable information, nor do they raise copyright concerns. Based on the experiments we conducted, we did not find evidence of systematic bias against different genders, languages, or regions. To the best of our knowledge, the experiments reported in this paper don't raise ethical concerns.

### A.2  REPRODUCIBILITY STATEMENT

We are committed to ensuring the reproducibility of our results. All code, data, and preprocessing scripts will be released publicly after the anonymous review period. Detailed hyperparameters, hardware specifications, and other experimental settings are documented in Appendix B. Together, these materials should allow independent researchers to fully reproduce our results.

### A.3  LLM USAGE DISCLOSURE

Large language models (LLMs) were used in a limited capacity during this work. First, AI tools were employed for minor English polishing, such as improving grammar and selecting more accurate word usage to convey ideas precisely. Second, AI tools were occasionally used to assist in debugging code during experiments. All scientific ideas, experimental designs, and contributions remain the authors' own.

## B  EXPERIMENTAL AND IMPLEMENTATION DETAILS

This section provides supplementary details on our experimental setup, models, datasets, and implementation.

### B.1  MODEL SETUP

Our experimental setup involves a comprehensive suite of models, detailed in Table 4. A key principle of our design is the direct comparability between baseline and student models: they share the same architecture, training hyperparameters, and number of training samples. The only distinction lies in their supervisory signal—baselines learn from ground-truth labels, while students learn from a teacher's outputs.

To fully explore the characteristics of each distillation method, we deviate from the standard paradigm. Specifically, we investigate setups both with and without the cross-entropy loss term on ground-truth labels for student training. This is controlled by a weighting coefficient, $\alpha$. When $\alpha = 1$, students learn exclusively from the teacher's supervisory signal. This ensures that any observed performance gain in a dirty student is attributable purely to the laundered knowledge from the teacher. When $\alpha = 0.5$, the student learns from a balanced mix of the teacher's signal and the ground-truth labels.

For the sake of brevity, we primarily present the results from the Soft Forward distillation with $\alpha = 1$ in the main body of the paper. However, our experiments comprehensively cover all three core distillation methods: (i) soft-label distillation with forward KL divergence (SoftFwd), (ii) soft-label distillation with reverse KL divergence (SoftRev) (Amara et al., 2024), which uses an alternative divergence measure, and (iii) hard-label distillation (Hard) (Hinton et al., 2015). Each distillation method is evaluated in settings both with and without the Ground Truth Cross-Entropy loss. Detailed results across all these configurations can be found in appendix C.

| Model | Description | Training Data | Supervisory Signal | Loss Function | $\alpha$ |
|---|---|---|---|---|---|
| $B_{clean}$ | Clean Baseline | Clean | Ground-truth labels | $\mathcal{L}_{CE}(y, \sigma(z_m))$ | 0 |
| $B_{dirty}$ | Dirty Baseline | Contaminated | Ground-truth labels | $\mathcal{L}_{CE}(y, \sigma(z_m))$ | 0 |
| $S_{clean}^{hard}$ | Student from $T_{clean}$ (pure) | Clean | Hard labels from $T_{clean}$ | $\mathcal{L}_{CE}(\hat{y}_t, \sigma(z_s))$ | 1 |
| $S_{dirty}^{hard}$ | Student from $T_{dirty}$ (pure) | Clean | Hard labels from $T_{dirty}$ | $\mathcal{L}_{CE}(\hat{y}_t, \sigma(z_s))$ | 1 |
| $S_{clean}^{hard,mix}$ | Student from $T_{clean}$ (mixed) | Clean | GT + Hard labels from $T_{clean}$ | $0.5 \cdot \mathcal{L}_{CE}(y, \sigma(z_s)) + 0.5 \cdot \mathcal{L}_{CE}(\hat{y}_t, \sigma(z_s))$ | 0.5 |
| $S_{dirty}^{hard,mix}$ | Student from $T_{dirty}$ (mixed) | Clean | GT + Hard labels from $T_{dirty}$ | $0.5 \cdot \mathcal{L}_{CE}(y, \sigma(z_s)) + 0.5 \cdot \mathcal{L}_{CE}(\hat{y}_t, \sigma(z_s))$ | 0.5 |
| $S_{clean}^{soft,fwd}$ | Student from $T_{clean}$ (pure) | Clean | Soft labels from $T_{clean}$ | $\mathrm{KL}(\sigma(z_t/\tau) \parallel \sigma(z_s/\tau))$ | 1 |
| $S_{dirty}^{soft,fwd}$ | Student from $T_{dirty}$ (pure) | Clean | Soft labels from $T_{dirty}$ | $\mathrm{KL}(\sigma(z_t/\tau) \parallel \sigma(z_s/\tau))$ | 1 |
| $S_{clean}^{soft,fwd,mix}$ | Student from $T_{clean}$ (mixed) | Clean | GT + Soft labels from $T_{clean}$ | $0.5 \cdot \mathcal{L}_{CE}(y, \sigma(z_s)) + 0.5 \cdot \mathrm{KL}(\sigma(z_t/\tau) \parallel \sigma(z_s/\tau))$ | 0.5 |
| $S_{dirty}^{soft,fwd,mix}$ | Student from $T_{dirty}$ (mixed) | Clean | GT + Soft labels from $T_{dirty}$ | $0.5 \cdot \mathcal{L}_{CE}(y, \sigma(z_s)) + 0.5 \cdot \mathrm{KL}(\sigma(z_t/\tau) \parallel \sigma(z_s/\tau))$ | 0.5 |
| $S_{clean}^{soft,rev}$ | Student from $T_{clean}$ (pure) | Clean | Soft labels from $T_{clean}$ | $\mathrm{KL}(\sigma(z_s/\tau) \parallel \sigma(z_t/\tau))$ | 1 |
| $S_{dirty}^{soft,rev}$ | Student from $T_{dirty}$ (pure) | Clean | Soft labels from $T_{dirty}$ | $\mathrm{KL}(\sigma(z_s/\tau) \parallel \sigma(z_t/\tau))$ | 1 |
| $S_{clean}^{soft,rev,mix}$ | Student from $T_{clean}$ (mixed) | Clean | GT + Soft labels from $T_{clean}$ | $0.5 \cdot \mathcal{L}_{CE}(y, \sigma(z_s)) + 0.5 \cdot \mathrm{KL}(\sigma(z_s/\tau) \parallel \sigma(z_t/\tau))$ | 0.5 |
| $S_{dirty}^{soft,rev,mix}$ | Student from $T_{dirty}$ (mixed) | Clean | GT + Soft labels from $T_{dirty}$ | $0.5 \cdot \mathcal{L}_{CE}(y, \sigma(z_s)) + 0.5 \cdot \mathrm{KL}(\sigma(z_s/\tau) \parallel \sigma(z_t/\tau))$ | 0.5 |

Table 4: Overview of the models in our experimental setup. $T_{clean}$ and $T_{dirty}$ denote the clean and dirty teachers. $z_m, z_s, z_t$ are the logits from the main model, student, and teacher, respectively. $\sigma$ is the softmax function, $y$ is the ground-truth label, $\hat{y}_t$ is the teacher's hard prediction, and $\tau$ is the temperature. Hard distillation uses Cross-Entropy (CE) loss. Soft distillation uses KL Divergence; forward KL is "mean-seeking," while reverse KL is "mode-seeking" (Wu et al., 2025). The $\alpha$ parameter controls the weight of the ground-truth loss term; $\alpha = 1$ indicates pure distillation, while $\alpha = 0.5$ indicates a mixed objective.

## B.2 DATASETS

We use eight public classification benchmarks, detailed in Table 5. We adopt BERT-base-uncased for teachers and DistilBERT-base-uncased for students/baselines.

| Benchmark | Task Type | Classes | Original Train Size | Original Test Size | Train Subset Ratio | Effective Train/Test Ratio |
|---|---|---|---|---|---|---|
| 20newsgroups | Topic Classification | 20 | 11,314 | 7,532 | 1.0 | 1.50 |
| agnews | Topic Classification | 4 | 120,000 | 7,600 | 0.1 | 1.58 |
| banking77 | Intent Classification | 77 | 10,003 | 3,080 | 1.0 | 3.25 |
| emotion | Emotion Classification | 6 | 16,000 | 2,000 | 1.0 | 8.00 |
| imdb | Sentiment Analysis | 2 | 25,000 | 25,000 | 1.0 | 1.00 |
| rotten_tomatoes | Sentiment Analysis | 2 | 8,530 | 1,066 | 1.0 | 8.00 |
| snli | Natural Language Inference | 3 | 550,152 | 9,824 | 0.1 | 5.60 |
| tweet_sentiment | Sentiment Analysis | 3 | 45,615 | 12,284 | 0.5 | 1.86 |

Table 5: Details of the benchmark datasets used in our experiments. The Train Subset Ratio adjusts the training set size to control the relative influence of training versus injected test data.

| Benchmark | Classes | Original Train Size | Size per Stratified Quintile | Original Test Size | Effective Train/Test Ratio |
|---|---|---|---|---|---|
| emotion | 6 | 16,000 | 3,200 | 2,000 | 1.60 |
| rotten_tomatoes | 2 | 8,530 | 1,706 | 1,066 | 1.60 |

Table 6: Dataset details for the controlled distribution gap experiments. The training set for each benchmark was partitioned into five stratified quintiles based on semantic similarity to the test set. The table shows the resulting size of each quintile and the corresponding effective train/test ratio.

## B.3 HYPERPARAMETERS AND COMPUTATIONAL RESOURCES

The same set of hyperparameters was used for training all baseline, teacher, and student models to ensure a fair and controlled comparison. We trained all models for a fixed number of epochs and did not use a development set for early stopping. The specific hyperparameters are detailed in Table 7.

We utilized NVIDIA A100 GPUs for the standard BERT-based experiments in Section 4. The LLM distillation experiments, as well as the controlled distribution gap studies in Section 5, were conducted on AMD MI250X GPUs.

| Hyperparameter | Value |
|---|---|
| Learning Rate | 2e-5 |
| Batch Size | 32 |
| Training Epochs | 3 |
| Distillation Temperature | 2.0 (for soft distillation only) |
| Max Sequence Length | 128 (512 for IMDB & 20newsgroups) |
| Random Seeds | 1, 42, 86, 358, 1024 |

Table 7: Hyperparameters for all model training.

### B.4 SIGNIFICANCE TESTING DETAILS

To ensure the reliability of our findings, we employed bootstrapping-based statistical tests. The detailed procedures for assessing the significance of accuracy gains and correlation coefficients are outlined below.

**Accuracy Gains (Clean vs. Dirty)**   To determine if the accuracy of a "dirty" model was significantly higher than its "clean" counterpart, we used a one-sided paired bootstrap test. For each of the five random seeds, we first drew 10,000 bootstrap samples (with replacement) from the test set predictions of the clean and dirty models. Then, for each bootstrap sample, we calculated the difference in accuracy between the dirty and clean models. Finally, the p-value was estimated as the proportion of bootstrap samples where the clean model's accuracy was greater than or equal to the dirty model's accuracy. To maintain a conservative assessment, we report the maximum p-value observed across the five random seeds for each benchmark comparison. In addition to significance testing, we quantified the uncertainty of the performance gains. We computed 95% confidence intervals (CI) for the accuracy difference using the bootstrap percentile method. Specifically, the lower and upper bounds of the CI correspond to the 2.5th and 97.5th percentiles, respectively, of the distribution of accuracy differences obtained from the 10,000 bootstrap samples. In our results, we report the average 95% CI across the five random seeds to represent the stability of the observed gains.

**Correlation Coefficients**   To confirm the stability and significance of the Pearson correlation coefficient between the laundering effect and the contamination effect, we used a two-sided bootstrap test. For each random seed, we performed 10,000 bootstrap resamples of the test set samples. For each resample, we re-calculated the Pearson correlation. The p-value was then derived from the distribution of these bootstrapped correlation coefficients to test the null hypothesis that the true correlation is zero. We report the maximum p-value across the five seeds.

### B.5 MODEL-AGNOSTIC DATA CHARACTERISTICS

To characterize the relationship between the training and test sets in a model-agnostic way, we compute a suite of similarity metrics. These metrics, all normalized to the range $[0, 1]$, help identify inherent data properties that might make a benchmark more susceptible to leakage. Let $\mathcal{C}_{\text{train}}$ and $\mathcal{C}_{\text{test}}$ denote the training and test corpora, respectively. For some metrics, the similarity is a direct comparison between corpora, while for others, it is an aggregation of sample-level calculations.

- **Jaccard Similarity**: Measures lexical overlap based on the set of unique n-grams ($N_g$) present in each corpus. This is a direct corpus-level comparison.

$$\text{Jaccard}(\mathcal{C}_{\text{train}}, \mathcal{C}_{\text{test}}) = \frac{|N_g(\mathcal{C}_{\text{train}}) \cap N_g(\mathcal{C}_{\text{test}})|}{|N_g(\mathcal{C}_{\text{train}}) \cup N_g(\mathcal{C}_{\text{test}})|}$$

- **TF-IDF Cosine Similarity**: Measures similarity by comparing the centroids of the corpora in the TF-IDF vector space. First, each corpus $\mathcal{C}$ is represented by its mean TF-IDF vector, $\vec{v}_{\text{tfidf}}(\mathcal{C})$, which is an aggregation of individual sample vectors.

$$\vec{v}_{\text{tfidf}}(\mathcal{C}) = \frac{1}{|\mathcal{C}|} \sum_{x \in \mathcal{C}} \vec{v}_{\text{tfidf}}(x)$$

The final similarity is the cosine distance between these two mean vectors.

$$\text{Sim}_{\text{TF-IDF}}(\mathcal{C}_{\text{train}}, \mathcal{C}_{\text{test}}) = \cos\_\text{sim}(\vec{v}_{\text{tfidf}}(\mathcal{C}_{\text{train}}), \vec{v}_{\text{tfidf}}(\mathcal{C}_{\text{test}}))$$

- **Average Embedding Similarity**: Similar to TF-IDF, this metric computes the cosine similarity between the mean Sentence-BERT embedding vectors of the corpora. The mean embedding vector for a corpus, $\vec{e}(\mathcal{C})$, is derived by averaging the embeddings of all its samples.

$$\vec{e}(\mathcal{C}) = \frac{1}{|\mathcal{C}|} \sum_{x \in \mathcal{C}} \text{emb}(x)$$

The similarity is then calculated between these two corpus-level representations.

$$\text{Sim}_{\text{AvgEmb}}(\mathcal{C}_{\text{train}}, \mathcal{C}_{\text{test}}) = \text{cos\_sim}(\vec{e}(\mathcal{C}_{\text{train}}), \vec{e}(\mathcal{C}_{\text{test}}))$$

- **Average Max Semantic Similarity**: Quantifies how well each test sample is represented in the training set. This metric is explicitly an aggregation of sample-level scores. For each test sample $x_{\text{test}}$, we find its highest cosine similarity to any sample in the training set, and then average these maximum similarity scores.

$$\text{Sim}_{\text{AvgMax}} = \frac{1}{|\mathcal{C}_{\text{test}}|} \sum_{x_{\text{test}} \in \mathcal{C}_{\text{test}}} \left( \max_{x_{\text{train}} \in \mathcal{C}_{\text{train}}} \text{cos\_sim}(\text{emb}(x_{\text{test}}), \text{emb}(x_{\text{train}})) \right)$$

- **Average Pattern Conformity**: Assesses how well test samples align with the dominant semantic patterns of the training set. We first run k-Means on the training embeddings to find $k$ centroids $\{c_i\}_{i=1}^{k}$. The metric is the average of each test sample's maximum cosine similarity to any of these centroids, making it a clear aggregation of sample-level conformity scores.

$$\text{PatternConformity} = \frac{1}{|\mathcal{C}_{\text{test}}|} \sum_{x_{\text{test}} \in \mathcal{C}_{\text{test}}} \left( \max_{i \in \{1, \ldots, k\}} \text{cos\_sim}(\text{emb}(x_{\text{test}}), c_i) \right)$$

### B.6 Verification of Controlled Distributional Gaps

To verify that our stratified splitting method effectively created controlled distributional gaps, we visualized the similarity between each training data level and the test set. Figure 5 shows the distribution of cosine similarities, confirming a systematic shift where Level 1 is most similar to the test set and Level 5 is least similar. Figure 6 further corroborates this by demonstrating a monotonic decrease across five different lexical and semantic similarity metrics, validating the integrity of our experimental setup.

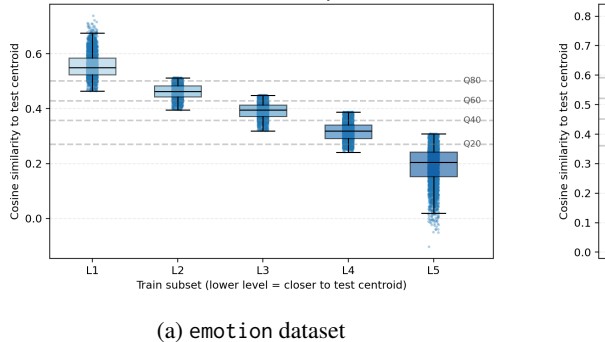 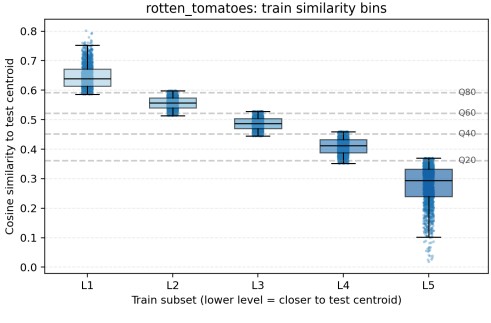

(a) emotion dataset        (b) rotten_tomatoes dataset

Figure 5: Similarity bins for the emotion (left) and rotten_tomatoes (right) datasets. Both plots show the distribution of cosine similarities between training samples in each level and the test set centroid, confirming that Level 1 is most similar and Level 5 is least similar. It is worth noting that the levels are not perfectly discrete, which is a natural consequence of our stratified splitting strategy, as samples are partitioned within each class before being aggregated into global levels.

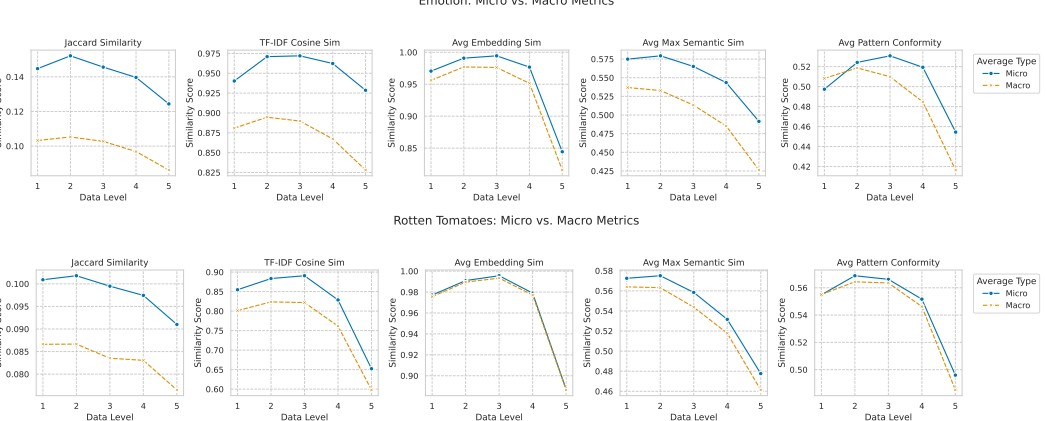

Figure 6: Verification of controlled distributional gaps using five similarity metrics for the `emotion` (top panel) and `rotten_tomatoes` (bottom panel) datasets. Both panels show a consistent decline across all metrics as the level increases from 1 to 5, validating the successful creation of a widening train-test gap.

## C    DETAILED EVALUATION STATISTICS AND SANITY CHECKS

To further validate the robustness of our main findings, we conducted a series of auxiliary experiments. These checks were designed to test our conclusions against alternative methodological choices and potential confounding factors, ensuring that the observed decontamination effect of knowledge distillation is a genuine and reliable phenomenon. This appendix details the methodology and results of these investigations.

### C.1    FULL PERFORMANCE DATA FOR BASELINE, TEACHER, AND STUDENT MODELS

| Benchmark | $T_{clean}$ | $T_{dirty}$ | $\Delta Acc_T$ | 95% CI | $B_{clean}$ | $B_{dirty}$ | $\Delta Acc_B$ | 95% CI | $S_{clean}^{soft,fwd}$ | $S_{dirty}^{soft,fwd}$ | $\Delta Acc_S$ | 95% CI |
|---|---|---|---|---|---|---|---|---|---|---|---|---|
| 20Newsgroups | 70.37 | 84.62 | +14.25*** | [13.4, 15.1] | 69.38 | 81.29 | +11.91*** | [11.1, 12.7] | 68.60 | 70.02 | +1.42*** | [0.9, 2.0] |
| AG News | 91.89 | 99.25 | +7.36 *** | [6.8, 8.0] | 91.64 | 98.31 | +6.67 *** | [6.1, 7.3] | 91.59 | 92.24 | +0.65* | [0.3, 1.0] |
| Banking77 | 78.65 | 83.60 | +4.95 *** | [3.7, 6.2] | 76.97 | 82.29 | +5.32 *** | [4.1, 6.6] | 62.21 | 64.44 | +2.23 | [0.6, 3.9] |
| Emotion | 92.29 | 98.35 | +6.06 *** | [5.0, 7.1] | 92.70 | 97.59 | +4.89 *** | [3.9, 5.9] | 93.10 | 93.76 | +0.66 | [-0.0, 1.4] |
| IMDb | 93.63 | 99.81 | +6.18 *** | [5.9, 6.5] | 92.78 | 99.65 | +6.87 *** | [6.6, 7.2] | 92.72 | 93.39 | +0.67*** | [0.5, 0.9] |
| Rotten Tomatoes | 85.14 | 98.65 | +13.51*** | [11.5, 15.6] | 84.20 | 97.52 | +13.32*** | [11.3, 15.4] | 83.77 | 84.33 | +0.56 | [-0.4, 1.6] |
| SNLI | 86.00 | 98.72 | +12.73*** | [12.1, 13.4] | 82.95 | 96.10 | +13.14*** | [12.5, 13.8] | 52.91 | 58.67 | +5.77 | [4.7, 6.8] |
| Tweet Sentiment | 67.81 | 96.82 | +29.01*** | [28.2, 29.8] | 67.33 | 92.99 | +25.66*** | [24.9, 26.5] | 66.99 | 70.24 | +3.25*** | [2.7, 3.8] |

Table 8: Detailed performance statistics using notation defined in Table 4. Values are percentage points. $\Delta Acc$ denotes the absolute performance gain from contamination (Dirty - Clean). Significance (*$p < 0.05$,** $p < 0.01$,*** $p < 0.001$) and 95% Confidence Intervals (CI) are reported.

Table 8 presents the complete dataset corresponding to the figure in the main body that illustrates the performance of the baseline, teacher, and student models. This table provides a detailed breakdown of the results.

### C.2    ROBUSTNESS ACROSS DISTILLATION STRATEGIES: THE PRIVACY-UTILITY TRADE-OFF

To address the question of whether specific distillation objectives facilitate or hinder data laundering, we extended our analysis to include Soft Forward, Soft Reverse, and Hard-label distillation, comparing "Pure" (teacher-only) signals against "Mixed" objectives (teacher + ground truth). The results are detailed in Tables 9, 10, and 11.

While our primary conclusion holds—that distillation generally acts as an effective decontaminator—a closer inspection reveals how different distillation strategies influence the risk of data laundering:

| Benchmark | $S_{clean}^{soft,fwd}$ | $S_{dirty}^{soft,fwd}$ | $\Delta Acc_S$ | 95% CI | $S_{clean}^{soft,fwd,mix}$ | $S_{dirty}^{soft,fwd,mix}$ | $\Delta Acc_S$ | 95% CI |
|---|---|---|---|---|---|---|---|---|
| 20Newsgroups | 68.60 | 70.02 | +1.42*** | [0.9, 2.0] | 69.04 | 70.13 | +1.09*** | [0.7, 1.5] |
| AG News | 91.59 | 92.24 | +0.65* | [0.3, 1.0] | 91.62 | 92.26 | +0.64** | [0.3, 1.0] |
| Banking77 | 62.21 | 64.44 | +2.23 | [0.6, 3.9] | 77.91 | 78.28 | +0.37 | [-0.4, 1.1] |
| Emotion | 93.10 | 93.76 | +0.66 | [-0.0, 1.4] | 92.86 | 93.17 | +0.31 | [-0.2, 0.8] |
| IMDb | 92.72 | 93.39 | +0.67*** | [0.5, 0.9] | 92.71 | 93.76 | +1.05*** | [0.9, 1.3] |
| Rotten Tomatoes | 83.77 | 84.33 | +0.56 | [-0.4, 1.6] | 84.07 | 84.65 | +0.58 | [-0.2, 1.4] |
| SNLI | 52.91 | 58.67 | +5.77 | [4.7, 6.8] | 68.52 | 74.18 | +5.66 | [4.8, 6.5] |
| Tweet Sentiment | 66.99 | 70.24 | +3.25*** | [2.7, 3.8] | 67.20 | 69.63 | +2.43*** | [2.0, 2.9] |

Table 9: Comparison of Student accuracy: **Soft Forward (Pure)** vs. **Soft Forward (Mixed)** distillation strategies.

| Benchmark | $S_{clean}^{soft,rev}$ | $S_{dirty}^{soft,rev}$ | $\Delta Acc_S$ | 95% CI | $S_{clean}^{soft,rev,mix}$ | $S_{dirty}^{soft,rev,mix}$ | $\Delta Acc_S$ | 95% CI |
|---|---|---|---|---|---|---|---|---|
| 20Newsgroups | 68.90 | 70.20 | +1.30 | [0.7, 1.9] | 69.16 | 70.19 | +1.03*** | [0.6, 1.4] |
| AG News | 91.47 | 92.09 | +0.63* | [0.2, 1.0] | 91.56 | 92.26 | +0.70** | [0.4, 1.0] |
| Banking77 | 61.68 | 63.34 | +1.66 | [0.0, 3.3] | 78.68 | 78.87 | +0.19 | [-0.5, 0.9] |
| Emotion | 93.11 | 93.48 | +0.37 | [-0.2, 1.0] | 92.87 | 93.25 | +0.38 | [-0.1, 0.9] |
| IMDb | 92.68 | 93.29 | +0.60*** | [0.4, 0.8] | 92.69 | 93.70 | +1.01*** | [0.8, 1.2] |
| Rotten Tomatoes | 83.71 | 84.09 | +0.38 | [-0.6, 1.4] | 84.15 | 84.28 | +0.13 | [-0.6, 0.9] |
| SNLI | 52.84 | 58.54 | +5.70 | [4.7, 6.7] | 65.84 | 71.76 | +5.91 | [5.0, 6.8] |
| Tweet Sentiment | 67.29 | 70.20 | +2.92*** | [2.4, 3.5] | 67.28 | 69.77 | +2.49*** | [2.0, 3.0] |

Table 10: Comparison of Student accuracy: **Soft Reverse (Pure)** vs. **Soft Reverse (Mixed)** distillation strategies.

| Benchmark | $S_{clean}^{hard}$ | $S_{dirty}^{hard}$ | $\Delta Acc_S$ | 95% CI | $S_{clean}^{hard,mix}$ | $S_{dirty}^{hard,mix}$ | $\Delta Acc_S$ | 95% CI |
|---|---|---|---|---|---|---|---|---|
| 20Newsgroups | 69.10 | 69.57 | +0.48 | [-0.0, 1.0] | 69.27 | 70.18 | +0.90** | [0.4, 1.4] |
| AG News | 91.72 | 91.80 | +0.08 | [-0.3, 0.5] | 91.66 | 92.16 | +0.49* | [0.1, 0.9] |
| Banking77 | 66.61 | 66.42 | −0.19 | [-1.4, 1.0] | 70.87 | 71.00 | +0.13 | [-1.1, 1.4] |
| Emotion | 92.88 | 93.57 | +0.69 | [-0.0, 1.5] | 92.90 | 93.34 | +0.44 | [-0.3, 1.2] |
| IMDb | 92.79 | 93.06 | +0.27 | [0.0, 0.5] | 92.82 | 93.58 | +0.76*** | [0.6, 1.0] |
| Rotten Tomatoes | 84.15 | 84.11 | −0.04 | [-1.0, 0.9] | 84.18 | 84.45 | +0.26 | [-0.8, 1.3] |
| SNLI | 52.69 | 58.17 | +5.47 | [4.4, 6.5] | 69.36 | 74.23 | +4.87 | [4.0, 5.7] |
| Tweet Sentiment | 67.16 | 69.33 | +2.17*** | [1.6, 2.7] | 67.26 | 69.01 | +1.75*** | [1.3, 2.3] |

Table 11: Comparison of Student accuracy: **Hard Label (Pure)** vs. **Hard Label (Mixed)** distillation strategies.

**1. Soft targets carry more risk than hard labels.** Across benchmarks, Student models trained on soft labels consistently exhibit larger performance gaps ($\Delta\text{Acc}_S$) compared to hard labels. This vulnerability is reflected in statistical significance. While Hard Pure distillation often yields insignificant results (indicating that no systematic laundering occurred), Soft methods frequently exhibit highly significant gaps (***, $p < 0.001$). We attribute this to the information richness of soft logits, which carry the teacher's full probability distribution, including the patterns learned from contamination. In contrast, Hard labels act as a filter, removing these subtle nuances and preventing systematic laundering.

**2. Mixed objectives can increase the visibility of data laundering.** The "Mixed" strategy, which combines the teacher's signal with the training data's ground-truth labels, tends to widen the performance gap ($\Delta\text{Acc}_S$). This effect is most pronounced in hard distillation (Table 11), where adding the ground truth often makes the laundering effect statistically significant. We hypothesize that the ground-truth signals may synergize with the signals from the contaminated teacher, making the laundering effect more pronounced in the student compared to when learning from the teacher alone. However, the exact mechanism behind this interaction requires further investigation.

**3. An Open Question: The Privacy-Utility Trade-off.** These findings highlight a dilemma rather than a simple solution. On one hand, using **Pure Hard Distillation** appears to be the safest method for decontamination, as it minimizes the transfer of test-set patterns. On the other hand, **Soft Mixed Distillation** provides significantly higher utility (e.g., Banking77 clean student accuracy increases from $\sim 66\%$ in Hard Pure to $\sim 78\%$ in Soft Mixed). Limiting the student to sparse information (Hard labels only) hampers learning efficiency. Therefore, how to balance the need for rich training signals against the risk of data laundering remains an open question for future research.

### C.3 Generalization Across Architectures Using State-of-the-Art LLM Teachers

To address the potential limitation of relying solely on BERT-based models and to validate the universality of our findings, we extended our analysis to include state-of-the-art lightweight Large Language Models (LLMs) as teachers. Specifically, we employed **Llama3.2-1B** and **Qwen3-0.6B**. These models represent the current state-of-the-art for their size class.

We maintained `distilbert-base-uncased` as the fixed student model to strictly isolate the influence of the teacher's architecture and capabilities. All experiments in this section utilize the **Soft Forward** distillation strategy ($\alpha = 1, \tau = 2.0$). Table 12 presents the detailed results, including 95% bootstrap confidence intervals (CI).

The evidence strongly supports our central claims. Despite employing highly capable teachers that memorize contaminated data extremely well, including perfect accuracy on Tweet Sentiment and Rotten Tomatoes, the behavior remains consistent. The laundering effect that reaches the student is still minimal. For instance, on `Tweet Sentiment`, while the Llama-3.2 teacher gains **25.80%** from direct contamination, the student's laundering gain is limited to just **1.91%**. The confidence intervals indicate that the laundering effect is statistically significant only for the Tweet Sentiment benchmark (its intervals excludes zero). Importantly, even in this one significant case, the estimated magnitude is still drastically reduced compared to the direct contamination gain. This pattern reinforces the role of Knowledge Distillation as a buffer to prevent the propagation of contamination from the teacher model.

### C.4 Pretraining Corpus Overlap Audit

A potential concern is that the base models used in our study, BERT and DistilBERT, might have been inadvertently exposed to benchmark test data in their original pretraining corpora. To ensure that our findings originate from our explicit contamination protocols rather than from such pre-existing issues, we conducted a corpus-level overlap audit. According to their respective documentation, both BERT and DistilBERT were pretrained on a combination of English Wikipedia and the BookCorpus (Zhu et al., 2015). We created a surrogate pretraining corpus composed of recent snapshot of Wikipedia

| Model | Benchmark | $T_{clean}$ | $T_{dirty}$ | $B_{clean}$ | $B_{dirty}$ | $\Delta Acc_B$ | 95% CI | $S^{soft,rev}_{clean}$ | $S^{soft,rev}_{dirty}$ | $\Delta Acc_S$ | 95% CI |
|---|---|---|---|---|---|---|---|---|---|---|---|
| LLAMA3.2-1B | Agnews | 92.58 | 99.98 | 91.09 | 98.32 | +7.23 *** | $[6.6, 7.8]$ | 91.08 | 91.46 | +0.39 | $[-0.01, 0.78]$ |
| | Emotion | 92.81 | 99.80 | 92.90 | 97.69 | +4.79 *** | $[3.9, 5.8]$ | 92.81 | 93.11 | +0.30 | $[-0.21, 0.82]$ |
| | Rotten Tomatoes | 90.66 | 100.00 | 84.11 | 97.56 | +13.45*** | $[11.4, 15.6]$ | 84.15 | 84.32 | +0.17 | $[-0.62, 0.98]$ |
| | Tweet Sentiment | 69.89 | 100.00 | 67.28 | 93.08 | +25.80*** | $[25.0, 26.6]$ | 67.13 | 69.04 | +1.91*** | $[1.35, 2.47]$ |
| QWEN3-0.6B | Agnews | 91.97 | 99.98 | 91.09 | 98.32 | +7.23 *** | $[6.7, 7.8]$ | 91.04 | 91.49 | +0.45 | $[0.08, 0.82]$ |
| | Emotion | 92.71 | 99.75 | 92.90 | 97.69 | +4.79 *** | $[3.8, 5.8]$ | 92.82 | 93.12 | +0.30 | $[-0.25, 0.86]$ |
| | Rotten Tomatoes | 87.15 | 100.00 | 84.11 | 97.56 | +13.45*** | $[11.4, 15.5]$ | 84.22 | 84.37 | +0.15 | $[-0.73, 1.03]$ |
| | Tweet Sentiment | 68.49 | 99.99 | 67.28 | 93.08 | +25.80*** | $[25.0, 26.6]$ | 67.10 | 69.10 | +2.00*** | $[1.43, 2.57]$ |

Table 12: Data Laundering results using modern LLMs (LLAMA3.2-1B and QWEN3-0.6B) as teachers. The Student model is fixed as DistilBERT trained with Soft Forward distillation. Gain denotes the absolute percentage point increase. Significance (***) indicates $p < 0.001$. 95% Confidence Intervals (CI) are computed via bootstrapping.

('20231101.en')[2] and the BookCorpusOpen[3]. Subsequently, we performed an exhaustive search to determine if any sentence from the test sets of our eight benchmarks appeared verbatim within this extensive corpus.

The audit revealed zero exact 13-gram matches between any of our benchmark test sets and the surrogate pretraining data. While this does not preclude more subtle forms of semantic overlap, it provides strong evidence that our results are not confounded by the most direct form of test set leakage into the pretraining pipeline of the models we used. This finding increases our confidence that the leakage effects studied in this paper are indeed a consequence of our controlled experiments.

---

[2] https://huggingface.co/datasets/wikimedia/wikipedia
[3] https://huggingface.co/datasets/lucadiliello/bookcorpusopen

