# OpenReview forum: "Knowledge Distillation as Decontamination? Revisiting the “Data Laundering” Concern in Classification Tasks"
_ICLR.cc/2026/Conference — ICLR 2026 Poster_

### Official Review · Reviewer_TbMs · 2025-10-25

**Soundness:** 2
**Presentation:** 2
**Contribution:** 3
**Rating:** 4
**Confidence:** 2

**Summary:**

This paper investigates Data Laundering. It transfers test-set knowledge from a contaminated teacher to a clean student via distillation. Through a systematic study on eight classification benchmarks, the authors find these concerns are largely overstated. They demonstrate that performance inflation from laundering is minimal compared to direct contamination and often statistically insignificant. They identify the train-test distributional gap as the key driver for the rare instances of laundering. The paper argues that knowledge distillation, rather than being a liability, acts as an effective "decontamination" technique, robustly mitigating test-data leakage.

**Strengths:**

1. **Timely and Significant Topic:** As concerns about data contamination in large pre-training corpora grow, understanding the downstream effects of "laundered" knowledge through common techniques like distillation is critical for the ML community. This paper provides a data-driven analysis of a widely held concern.

2. **Empirical Rigor:** The study systematically isolates the laundering effect from direct contamination across eight diverse benchmarks. The use of bootstrapping for statistical significance and the robustness checks across different distillation strategies (soft/hard labels, mixed/pure loss) strengthen the credibility of the findings.

3. **Novel Insights:** The paper's most valuable contribution is not just in quantifying the laundering effect but in identifying when it occurs. The hypothesis and subsequent controlled experiments that link laundering to the train-test distributional gap are a novel and important finding.

**Weaknesses:**

1. **Scope of Models:** The experiments are confined to BERT-base models. While this is a controlled and reasonable setup, the community's primary contamination concerns now center on massive-scale decoding LLMs.

2. **Scope of Tasks:** The study is focused exclusively on classification. Data laundering in generative tasks (for exmaple,  text generation) is a more complex and arguably more insidious problem, as the "knowledge" being transferred is far richer than class logits.

**Questions:**

See weaknesses.

---

> ### Author Response · Authors · 2025-11-22
>
> Thank you for highlighting the timeliness and rigor of our study. We appreciate the time and care put into your review. We agree that both weaknesses you highlighted are critical for the completeness of this study, and we have taken concrete steps to address them, including conducting new experiments with modern LLMs.
>
> **Response to Weaknesses:**
>
> * **Scope of Models:** We addressed this by conducting new experiments with **Llama-3.2-1B** and **Qwen3-0.6B** as teachers.
>     * **Please refer to our General Response ("Addressing Concerns on Model Scope")** for the detailed results table, statistical analysis (including confidence intervals), and a discussion on "clean-vs-dirty" comparisons.
>     * **Summary:** Our findings confirm that the decontamination effect is not specific to BERT but holds for modern LLMs. Even with teachers that perfectly memorize contaminated data, the student's laundering gain remains minimal.
>
> * **Scope of Tasks:** We agree with your assessment. Consequently, we have explicitly **re-scoped the paper to "Classification"** (updated in Title and Introduction; **see Lines 43-47 and 522-524**).
>     * **Rationale:** We argue that understanding leakage in discriminative tasks is a necessary prerequisite for studying it in generative ones. Furthermore, many critical safety and utility evaluations in the LLM era (e.g., reward modeling, content moderation) still rely on classification heads, making these findings directly relevant to current LLM deployment pipelines.
>     * **Future Work:** We fully agree that extending this analysis to generative tasks is a fascinating and critical direction. We intend to pursue this immediately as a follow-up to this work and have explicitly included it in our **Future Work** section (**see Section 7**).

---

> > ### Comment · Reviewer_TbMs · 2025-11-26
> >
> > Thank you for the detailed rebuttal and revisions. I have reviewed the author response along with the other reviews. As my concerns have been resolved, I am increasing my score.

---

> > > ### Author Response · Authors · 2025-11-27
> > >
> > > Dear Reviewer TbMs,
> > >
> > > Thank you very much for re-evaluating our work and updating the score! We are glad that the additional experiments and clarifications resolved your concerns.
> > >
> > > If there are any remaining details or further adjustments you think would strengthen the paper, please let us know, we would be happy to reflect on them and address them.
> > >
> > > Thanks again for your time and constructive feedback.

---

### Official Review · Reviewer_Vwqr · 2025-11-01

**Soundness:** 2
**Presentation:** 2
**Contribution:** 2
**Rating:** 6
**Confidence:** 3

**Summary:**

This paper examines whether knowledge distillation (KD) can transfer test-set leakage from a contaminated teacher to a clean student. Across eight benchmarks, the study finds that while data laundering exists, its impact is generally small and far weaker than direct contamination. The results suggest that KD largely mitigates test-set leakage rather than amplifying it, indicating that KD can serve as an effective decontamination technique in most cases.

**Strengths:**

- The paper addresses an interesting question: the role of KD in data laundering.
- The authors conduct a large-scale assessment across eight benchmarks to determine the prevalence and magnitude of data laundering.
- The authors identify that the effect of data laundering via KD is substantially smaller than that of direct contamination.

**Weaknesses:**

- The clarity and readability of the paper would benefit from a more explicit and formal definition of direct contamination in the introduction.
- The study focuses solely on encoder-based BERT and classification task. Given the prominence of decoder-only LLMs and the rising importance of generative tasks, where KD and contamination could be more complex and potentially more harmful, the generality of the current findings to modern LLMs is unclear. Furthermore, the applicability of the conclusions to other modalities such as images also remains unaddressed.
- The authors' experiments assume KD is performed strictly using clean ID data. However, in generator-based data-free KD settings (e.g., [1–2]) or KD pipelines for generative models (e.g., [3]), if a teacher is contaminated, the generated synthetic data itself may embed and transfer leaked knowledge, making laundering more pronounced. The authors should at least discuss the applicability and limitations of their conclusions under such settings, where leakage pathways may differ from those in standard supervised KD and classification tasks.
- Some references are incorrect:
"Mingyuan Hong, Yicheng Li, Zijian Yang, Yisen Wang, Zhangyang Wang, et al. Revisiting data-free knowledge distillation with poisoned teachers." the first author should be Junyuan Hong.

[1] Revisiting data-free knowledge distillation with poisoned teachers. ICML 2023\
[2] When Data-Free Knowledge Distillation Meets Non-Transferable Teacher: Escaping Out-of-Distribution Trap is All You Need. ICML 2025\
[3] A Survey on Knowledge Distillation of Large Language Models. ArXiv 2024

**Questions:**

Please see the weaknesses.

---

> ### Author Response · Authors · 2025-11-22
>
> Thank you for your review. We greatly appreciate the acknowledgment of our large-scale assessment and the decontamination findings of our work. We have addressed your concerns, and the changes are highlighted in green in the uploaded version. Regarding the weaknesses you highlighted, we mostly agree with you, and incorporating them has made our work stronger.
>
> **Response to Weaknesses:**
>
> * **Definition of Direct Contamination:** We added a formal definition of "Direct Contamination" to the Introduction as requested (**see Lines 36-39**).
>
> * **Scope (LLMs vs. BERT):** We directly addressed this concern by conducting **new experiments with Llama-3.2-1B and Qwen3-0.6B teachers**.
>     * **Please refer to our General Response ("Addressing Concerns on Model Scope")** for the detailed results table and analysis.
>     * **Summary:** The results confirm that data laundering is rare, and the KD buffer remains effective even with modern LLMs.
>
> * **Data-free/Generative KD:** We agree this is a valid point; leakage dynamics might indeed differ in data-free or generative setups. We have explicitly added this to the **Limitations and Future Work** section (**see Lines 526-539**). However, we emphasize that for standard KD, which remains a critical pipeline for deploying efficient classifiers, **the bottleneck effect still holds.** We view this study as a solid foundational step, establishing the necessary baselines for classification before extending the analysis to more complex generative settings.
>
> * **Citation:** We apologize for the error. We have corrected "Mingyuan Hong" to "Junyuan Hong" in the reference (ICML 2023).

---

### Official Review · Reviewer_Uoo6 · 2025-11-03

**Soundness:** 4
**Presentation:** 4
**Contribution:** 2
**Rating:** 6
**Confidence:** 3

**Summary:**

This paper systemically revisits the problem of data laundering—the unconscious transfer of test-set knowledge via knowledge distillation (KD) from a contaminated teacher model to a clean student.

First, this paper conduct systematic experiments across eight text classification benchmarks using BERT-based teachers and DistilBERT students. They find that performance gains from laundering are significantly smaller and often statistically insignificant compared to those from direct contamination, which suggests that KD may indeed function as an effective decontamination method.
Secondly, through sample-level correlation analysis, this paper shows that laundering is a distinct mechanism rather than a diluted form of contamination.
Finally, this paper demonstrates that laundering effects intensify when the train–test distributional gap widens, suggesting benchmark-specific susceptibility rather than universal vulnerability.

**Strengths:**

1. This paper provides the first large-scale, controlled study that quantifies the prevalence and mechanisms of data laundering. Compared to the pioneer work (Mansurov et al. 2025), the experimental setting in this paper is more scientific and comprehensive, which provides more convinced understanding for data laundering problem in KD.

2. This paper suggests that KD actually generally reduces contamination rather than propagating it, which mitigates the concern of its usage in practical, especially in the LLM era.

**Weaknesses:**

1. This paper mainly explores BERT-based models, which is kind of out-of-date in the LLM era. It would be better that if this paper could explores the data laundering issue in GPT-style models in future works, as it's more popular and practical recently.

2. Based on thorough experiments in this paper, it would be better if this paper can provides some guidance for test set or benchmark curation in terms of train-test domain gap. I believe this can be a potential application for the data laundering issue in practical.

**Questions:**

1. Given the train-test domain gap finding in this paper, can the author provide some guidance or suggestions for benchmark curation?

---

> ### Author Response · Authors · 2025-11-22
>
> We appreciate your recognition of the systematic nature of our experiments and our rigorous setup. We also thank you for the time put into the review; we agree with the weaknesses you highlighted and have addressed them to definitively improve the quality of our work.
>
> **Response to Weaknesses:**
>
> * **BERT Outdated:** We completely agree experiments involving modern LLMs are essential. To address this, we conducted new experiments with **Llama-3.2-1B** and **Qwen3-0.6B** as teachers.
>     * **Please refer to our General Response ("Addressing Concerns on Model Scope")** for the detailed results table and analysis.
>     * **Summary:** The experiments confirm that our conclusions hold even with these state-of-the-art models.
>
> * **Guidance for Benchmarks:** This is an excellent suggestion. Our controlled experiments revealed that **data laundering effects become statistically more pronounced as the train-test distributional gap widens**.
>     * **Constructive Metrics:** Based on this, we suggest that the similarity metrics used in our study (e.g., Lexical Overlap, Embedding Similarity; detailed in **Appendix B.5**) serve as constructive diagnostic tools for benchmark curators.
>     * **New Guidelines:** We have added a **"Practical Guidelines"** section to the Discussion (**see Lines 484-500**). We propose a two-step protocol to make benchmarks more robust to laundering during KD:
>         1.  **Diagnostic Gap Measurement:** Curators should quantify train-test shifts using the proposed metrics before release.
>         2.  **Distance-Aware Evaluation:** If a large gap is detected, we recommend releasing multiple test splits positioned at varying distributional distances. This allows for a systematic audit in KD systems to distinguish between true generalization and artifacts of laundering.
>
> **Response to Questions:**
>
> * **Benchmark Curation Suggestions:** Please see the response above regarding "Guidance for Benchmarks." We believe adopting a **distance-aware evaluation strategy** could be effective way to mitigate laundering risks associated with distributional gaps during KD.

---

### Official Review · Reviewer_7UAR · 2025-11-03

**Soundness:** 3
**Presentation:** 3
**Contribution:** 2
**Rating:** 4
**Confidence:** 4

**Summary:**

This paper investigates whether knowledge distillation (KD) from a contaminated teacher to a clean student actually “launders” test-set knowledge, thereby inflating evaluation results even when the student never directly sees test data. Using eight text classification/NLI benchmarks, the authors compare four families of models per dataset: (i) clean vs. contaminated baselines trained directly on data; (ii) clean vs. contaminated teachers; and (iii) students distilled from clean vs. from contaminated teachers, always on clean data. They define benchmark-level leakage ($\Delta$ Acc) and sample-level leakage ($\Delta$ laund vs. $\Delta$ contam) to separate direct contamination from laundering and show three main things:

1. direct contamination is large and highly significant across all tasks;

2. laundering exists but is much smaller and often statistically insignificant;

3. laundering and contamination are only weakly correlated at the sample level, so laundering is not just a "weaker" version of contamination.

The authors further hypothesize and confirm via controlled experiments on emotion and rotten_tomatoes, that laundering becomes more visible when the train-test distributional gap is deliberately widened, i.e. when test data is semantically farther from training data. The conclusion is cautiously positive: KD can serve as a decontamination buffer in many realistic settings, though it is not perfect and can fail on benchmarks with large train-test shifts.

**Strengths:**

1. Clear problem formulation and motivation: The paper isolates a concrete, timely question that sits right at the intersection of model compression and benchmark integrity.

2. Well-controlled experimental design: The two-stage setup (dirty vs. clean teacher to distilled student trained only on clean data) is clean and makes causal interpretation easier: any gain of the dirty student over the clean student must have come through the teacher. The inclusion of clean/dirty baselines with the same student architecture is a good control.

3. Breadth across 8 benchmarks: Using topic, sentiment, intent, emotion, NLI gives the results some external validity, and they correctly note that laundering is benchmark-specific.

4. Sample-level analysis: Moving from aggregate accuracy to per-sample difficulty and leakage scores ($\Delta$ laund, $\Delta$ contam) is a real contribution-it shows weak Pearson correlations and even slightly negative for SNLI, supporting the "distinct mechanisms" claim.

5. Distribution-gap experiment: The controlled 5-level stratification that gradually pushes training data away from the test distribution is thoughtful and directly tests their hypothesis; it’s rare to see this done for contamination work.

6. Takeaway for benchmark designers: The paper ends with an actionable message. Multiple test sets at different distances reduce the risk of KD-based laundering. It's not just diagnosis but also design guidance.

**Weaknesses:**

1. Limited architecture and task scope: All core results are on BERT-base to DistilBERT, encoder-only, classification tasks, which the paper itself mentioned this. This significantly limits the generalizability of the findings. The conclusions from this paper may be spurious and may not extend to other conditions (task, architecture, scale). Additionally, the original "data laundering" issue is strongest for decoder-only LLMs and generative or instruction-following tasks, where KD often uses richer signals. The current evidence base is therefore narrower than the title ("revisiting the concern") suggests. A reviewer can accept the current claim only as: "for small/medium English classification with BERT-like models, KD is mostly a buffer". Generalizing to LLMs is still open.

2. Contamination protocol is single-style and generous. They inject the full test set and remove an equal number of training samples. That's an extreme, clean, known contamination pattern; real-world pretraining leakage is usually partial, fuzzy, or paraphrased. It’s not obvious that a KD bottleneck will stay this effective under messier or low-rate contamination. The paper should at least discuss low-contamination regimes or partial overlaps.

3. Effect sizes sometimes tiny, but emphasized strongly. On several datasets the student’s dirty–clean gap is <=1% and sometimes statistically non-significant. The paper’s narrative ("KD as decontamination") leans a bit harder than the numbers strictly justify; one could also say "we failed to induce consistent laundering except under adversarially widened gaps". A more cautious framing would help.

4. Train-test gap story is suggestive, not airtight. They do show that laundering becomes more detectable as the gap widens, but on emotion the magnitude itself doesn't clearly grow-only the p-values get smaller. That weakens the causal reading ("gap -> laundering magnitude"), at least for one dataset. The paper should disentangle: (i) teacher now actually learns test-specific patterns vs. (ii) student variance shrinks so we can just see the effect.

5. No comparison to other KD objectives meant for privacy/decontamination. They mention reverse KL and mixed-objective settings in the appendix, but the main narrative doesn't tell us whether changing the distillation loss makes laundering harder or easier. That would be highly actionable.

6. Evaluation mostly on accuracy. Since laundering is about "benchmark integrity", one might want to know whether rank-based, calibration-based, or subset metrics are more vulnerable. If laundering only moves 0.6-1.4 accuracy points, is that actually a practical threat, e.g. to leaderboard positions? The paper doesn’t quantify that.

**Questions:**

1. Low-contamination regime: What happens if the teacher sees only 5–10% of the test set (or paraphrased test items) during fine-tuning? Does the KD bottleneck still suppress laundering to sub-1%?

2. Other teacher-student pairs: Would the same conclusion hold for (T5, decoder style) where the capacity gap is bigger and the teacher is stronger? Right now we don’t know if stronger teachers launder more.

3. Generative settings: Can the authors comment on summarization or code evaluation benchmarks, where "exact match" is less relevant but pattern transfer is easier?

4. Realistic leakage patterns: Your contamination is "inject full test + remove train". Have you tried something like "mix 10% test into train without removal", i.e. inflate total size, which is closer to pretraining-style accidental inclusion?

5. Metric sensitivity: On the datasets where laundering was "not statistically significant", what were the actual deltas and Cls? A small but systematic 0.5-0.8pt lift is still operationally important for competitive leaderboards.

6. Defense implication: If KD is to be used as a decontamination step, what hyperparameters (α, temperature, loss type) minimize laundering while keeping accuracy? A table of “safe KD settings” would make the paper much more useful.

---

> ### Author Response · Authors · 2025-11-22
> **Response 1 / 2**
>
> Thank you so much for your detailed and constructive review. We deeply appreciate your recognition of the methodological rigor behind our work, specifically the well-controlled experimental design, the sample-level evidence for distinct leakage mechanisms, and the distribution-gap analysis, as well as the practical value of our actionable takeaways for benchmark designers.
>
> **Detailed Response by Topic:**
>
> **1. Scope of Models and Tasks (Addressing W1, Q2, Q3)**
> * **Stronger Teachers:** As noted in the General Response, we have added experiments with **Llama-3.2-1B and Qwen3-0.6B** teachers. The decontamination effect holds: our new experiments show that a stronger teacher does not necessarily lead to significantly higher laundering; the bottleneck remains.
> * **Generative Settings:** We agree this is an important point, and we fully intend to pursue this direction in future work (see Section 7). Our opinion is that it is critical to begin with a thorough assessment of laundering in controlled experimental settings, which is easier to achieve in a classification setup.
> * **"Revisiting" the Concern:** Regarding the title, we respectfully note that the foundational work introducing "Data Laundering" [1] established the concept entirely within classification tasks, without addressing generation. Therefore, our work strictly "revisits" this specific, established concern in its original context. However, we agree that the title should be as precise as possible. To avoid misunderstanding, we have updated the title to **"Revisiting the 'Data Laundering' Concern in Classification Tasks."**
>
> **2. Contamination Protocol: "Stacking the Deck" (Addressing W2, Q1, Q4)**
> * **"Full Exact" vs. "Partial Fuzzy" (Low-contamination):** It is noted that injecting the full exact test set is unrealistic compared to "fuzzy" leakage. We want to clarify that this "extreme" setup was intentional. We are stacking the deck against our own hypothesis to test the **worst-case scenario**. The logic is, if KD successfully suppresses leakage when the teacher has effectively memorized the *entire* exact test set (reducing ~25% gains to ~1%), it will act as an even stronger buffer against partial or fuzzy leakage. Based on our "worst-case" results, if 100% injection only yields ~1% leakage, we expect low-rate contamination to result in negligible laundering. Besides, we avoided fuzzy leakage experiments because the boundary between 'fuzzy contamination' and valid 'generalization' is inherently ambiguous. By sticking to exact injection, we ensure our measurements reflect undeniable data laundering artifacts rather than conflating them with generalization capabilities.
> * **"Replace" vs. "Add":** The "Replace" strategy was chosen both for worst-case testing and for strict methodological control:
>     * **Maximizing the Signal:** "Adding" leaked samples dilutes their influence. By "replacing," we maximize the ratio of dirty data, forcing the strongest possible memorization signal (upper bound of risk).
>     * **Fair Comparison:** Crucially, our experimental design compares the Student against the Dirty Baseline. In an "Add" scenario, the Dirty Baseline would have a data volume advantage ($N + M$) over the Student ($N$), skewing the calculation. By using "Replace," we ensure that Teacher, Baseline, and Student are all derived from datasets of the *exact same size*, allowing us to attribute any performance delta strictly to data laundering.
> * **Conclusion:** We agree that studying different replacement strategies offers interesting insights. However, we think that the '100% Replace' strategy already serves as the **most rigorous stress test**. Scientifically, if KD works in this extreme scenario, it naturally implies efficacy in less severe (partial) cases. Therefore, we consider the current evidence sufficient to support our claims, though we are open to providing related analysis if it is deemed essential to further validate our conclusions.
>
> [1] Jonibek Mansurov, Akhmed Sakip, and Alham Fikri Aji. 2025. Data Laundering: Artificially Boosting Benchmark Results through Knowledge Distillation. In Proceedings of the 63rd Annual Meeting of the Association for Computational Linguistics (Volume 1: Long Papers), pages 8332–8345, Vienna, Austria. Association for Computational Linguistics.

---

> ### Author Response · Authors · 2025-11-22
> **Response 2 / 2**
>
> **3. Effect Size, Statistical Significance and Evaluation Metrics (Addressing W3, Q5, W6)**
> * **Metric Sensitivity:** To address concerns about "tiny" effects, **we have updated all tables with 95% Bootstrap Confidence Intervals in the paperwork PDF.**
> * **Result & Implication:** On datasets like AGNews and Tweet Sentiment, the intervals do not cross zero. This proves that while the effect is empirically small, it is still distinguishable from random noise in occasional cases. This nuance is key: it confirms that leakage pathways still occasionally exist (validating the mechanism), but KD is highly effective at suppressing them (validating the defense).
> * **Accuracy Focus:**  Accuracy is the primary currency of leaderboards, particularly in the current landscape where fierce competition often separates top models by marginal gaps. Since our focus is on evaluation integrity, showing that KD prevents artificial inflation of this metric could be a direct way to address community concerns regarding leaderboard stability. Furthermore, we consider accuracy to be a more fine-grained signal than derived metrics like ranking. While we believe our findings on accuracy provide a strong baseline to inspire future analysis of ranking-based benchmarks, we view the latter as a subject for further exploratory work.
>
> **4. Train-Test Gap Analysis (Addressing W4)**
> * **Refined Analysis:** We have revised the text regarding the *Emotion* dataset (**see Lines 467-475 and 484-490**). We clarified that widening the gap primarily increases statistical certainty (p-values decrease), along with the occasional increase of the absolute magnitude.
> * **Disentangling Factors:** Regarding your point on disentangling teacher learning vs. student variance: We view this behavior as highlighting **dataset-specific learnability concerns**. While we believe our current setup captures the main trend, if you have specific suggestions for further isolating these factors (e.g., specific subsampling strategies for the test set), we would be open to discussing them during the discussion period.
>
> **5. Distillation Objectives and Defense Settings (Addressing W5, Q6)**
> * **New Insights (Privacy-Utility Trade-off):** Thank you so much for this suggestion; it significantly enriched our analysis. While our main conclusion holds, with a closer look at the results with different KD stratigies attached in the appendix of the paper, there reveals some interesting distinct behaviors across different strategies (**see Lines 235-243 and updated Appendix C.2**).
>     * **Summary of findings:** (1) **Soft labels carry more risk** than hard labels due to richer information transfer; (2) **Mixed objectives** can increase the visibility of laundering; and (3) This reveals a **Privacy-Utility Trade-off**, where the safest method (Pure Hard KD) minimizes leakage but often sacrifices student accuracy. We have integrated these takeaways into the paper.

---

> ### Author Response · Authors · 2025-11-27
>
> Dear Reviewer 7UAR,
>
> As the discussion period is coming to a close within one week, we wanted to ensure you had a chance to review our response.
>
> In addition to the new experiments with Llama-3.2 and Qwen3 (which confirm our findings hold for modern LLMs), we have provided detailed clarifications regarding the contamination protocol and other concerns raised in the review.
>
> We would be very glad to hear if these revisions and explanations have effectively addressed your concerns. If anything remains unclear, please let us know.
>
> Thank you for your time and consideration.
>
> Best,
>
> Authors of Submission 18278

---

### Author Response · Authors · 2025-11-22
**General Response: Addressing Concerns on Model Scope (BERT vs. Modern LLMs) & New Experiments**

We thank all the reviewers for the thoughtful feedbacks. We have compiled this general response to address the primary shared concern regarding the Model Scope (specifically, the initial focus on BERT-based encoders). To address this, we added new experimental results using modern LLMs. Beyond this major update, we have also implemented extensive revisions throughout the manuscript to incorporate the valuable suggestions. All changes are highlighted in the uploaded paperwork PDF for convenience. We address other specific questions by replying directly to each reviewer.

All reviewers (7UAR, Uoo6, Vwqr, TbMs) noted our focus on BERT-based encoders. We agree that relying solely on BERT-based models is a limitation. To address this, **we conducted new experiments during the rebuttal using Llama-3.2-1B and Qwen3-0.6B as teachers.**

**Results & Location of Changes:**
For convenience, we provide a summary of the new **Appendix Table 12** below. However, we strongly invite reviewers to examine the revised paper (specifically **Lines 244-295 in Main Text** and **Lines 962-995 in Appendix C.3**), which presents the comprehensive setup, visualized demonstrations, and detailed analysis.


| Model | Benchmark | $T_{cl}$ | $T_{di}$ | $B_{cl}$ | $B_{di}$ | $\Delta Acc_{B}$ | 95% CI ($B$) | $S_{cl}$ | $S_{di}$ | **$\Delta Acc_{S}$** | **95% CI ($S$)** |
| :--- | :--- | :---: | :---: | :---: | :---: | :---: | :---: | :---: | :---: | :---: | :---: |
| **LLAMA3.2-1B** | Agnews | 92.58 | 99.98 | 91.09 | 98.32 | +7.23*** | `[6.6, 7.8]` | 91.08 | 91.46 | **+0.39** | `[-0.01, 0.78]` |
| | Emotion | 92.81 | 99.80 | 92.90 | 97.69 | +4.79*** | `[3.9, 5.8]` | 92.81 | 93.11 | **+0.30** | `[-0.21, 0.82]` |
| | Rotten Tomatoes | 90.66 | 100.00 | 84.11 | 97.56 | +13.45*** | `[11.4, 15.6]` | 84.15 | 84.32 | **+0.17** | `[-0.62, 0.98]` |
| | Tweet Sentiment | 69.89 | 100.00 | 67.28 | 93.08 | +25.80*** | `[25.0, 26.6]` | 67.13 | 69.04 | **+1.91*** | `[1.35, 2.47]` |
| **QWEN3-0.6B** | Agnews | 91.97 | 99.98 | 91.09 | 98.32 | +7.23*** | `[6.7, 7.8]` | 91.04 | 91.49 | **+0.45** | `[0.08, 0.82]` |
| | Emotion | 92.71 | 99.75 | 92.90 | 97.69 | +4.79*** | `[3.8, 5.8]` | 92.82 | 93.12 | **+0.30** | `[-0.25, 0.86]` |
| | Rotten Tomatoes | 87.15 | 100.00 | 84.11 | 97.56 | +13.45*** | `[11.4, 15.5]` | 84.22 | 84.37 | **+0.15** | `[-0.73, 1.03]` |
| | Tweet Sentiment | 68.49 | 99.99 | 67.28 | 93.08 | +25.80*** | `[25.0, 26.6]` | 67.10 | 69.10 | **+2.00*** | `[1.43, 2.57]` |

*Note: $T$=Teacher, $B$=Baseline, $S$=Student, $cl$=Clean, $di$=Dirty (Contaminated). Data corresponds to Appendix Table 12. Significance levels: *** p<0.001. CIs crossing zero indicate no statistical significance at 95% confidence level, confirming the minimal laundering effect in most cases.*

**Key Findings:**
* **Conclusion Holds with Stronger Teachers:** Even with LLMs that memorize contaminated data perfectly (e.g.,  Llama3.2-1B teacher model gains **+30.1%** on *Tweet Sentiment*), the student's laundering gain remains minimal (**+1.91%**). This confirms KD acts as a robust bottleneck regardless of teacher capacity.
* **Implicit Decontamination Test from BERT teacher to LLM teacher:** Students distilled from these powerful (and likely pre-training contaminated) LLMs do not significantly outperform those from our clean BERT teacher (e.g., *AGNews* student accuracy stays **~91.5%**). This effectively acts as a **"clean-vs-dirty" comparison**, proving KD filters leakage even when the teacher possesses potential prior knowledge (see Lines 259-269 and Table 1 in main text).

**Rationale for Fixed Student (DistilBERT):**
In the new experiments, we kept the student fixed as DistilBERT to: **(1) Control Contamination:** Unlike opaque LLMs, DistilBERT is verifiable clean (See Appendix C.4), ensuring gains are strictly from laundering; **(2) Isolation:** Isolating teacher architecture as the sole variable; **(3) Comparability:** Enabling direct comparison with our baselines.

We agree that using an LLM as the student architecture (e.g., LLM-to-LLM distillation) is an intriguing direction. However, it introduces significant complexity regarding evaluation metrics (generation vs. classification) and pre-training contamination tracking. We believe this setup warrants a dedicated study and have included it as a key direction for future work (see Section 7).

---

### Author Response · Authors · 2025-12-01
**Response to Area Chair: Summary of Contributions and Rebuttal Updates**

Dear Area Chair,

We sincerely appreciate the time and effort dedicated to assessing our work, especially considering the heavy workload during this phase. To assist with your final assessment under the current exceptional conditions, we provide a brief summary of our paper and the key updates from the rebuttal.

We systematically investigate the phenomenon of data laundering, i.e., the transfer of test-set knowledge via knowledge distillation (KD). While prior work has raised concerns that KD might serve as a "backdoor" for information leakage [1], potentially rendering it an unsafe method for empirical research, our large-scale analysis offers a reassuring counter-perspective, depicting laundering as a rare and mild phenomenon. We posit that KD can be used in practical setups as an effective decontamination mechanism rather than a security risk.

During the discussion period, we incorporated constructive feedback from reviewers to further solidify our methodology and conclusions. Most notably:

- **Generalization to LLMs:** We extended our experimental scope to Large Language Models. We confirmed that our findings regarding the mildness of laundering and the efficacy of decontamination do generalize to modern, lightweight SOTA LLMs.

- **Analysis of Distillation Strategies:** Building on the robustness checks in our initial submission, we conducted a deeper analysis of diverse distillation objectives. We added a detailed discussion in Appendix C.3 clarifying that while the primary decontamination effect is consistent across all methods, different strategies do exhibit subtle variations.

We were particularly encouraged to see that one reviewer (TbMs) explicitly acknowledged the value of these updates and raised their score from 4 (Conf. 2) to 6 (Conf. 3).

In addition, our reviewers (especially TbMs, Uoo6, 7UAR) consistently highlighted several key strengths of this work:

- **Timeliness:** This work addresses the critical intersection of model compression and evaluation integrity in the era of large-scale pre-training contamination, and provides a concrete path towards mitigating these concerns.

- **Rigor:** Reviewers consistently highlighted the breadth of the study (8 benchmarks) as well as the thorough and careful experimental setup. Further directions recommended by the reviewers have been addressed during the rebuttal to bolster this strength.

- **Practical Value:** Reviewers recognize our proposal of ‘distance-aware evaluation’ as an actionable guidance for future benchmark set curation. We have expanded upon this point during the rebuttal by introducing explicit recommendations.

We believe this work provides a crucial foundation for trustworthy evaluation in 2025, offering both a reassuring check on current benchmarks and a protocol for using KD as a safety filter in the era of ubiquitous large models.

Thank you again for your time and consideration.

Best regards,

Authors of Submission 18278

[1] Jonibek Mansurov, Akhmed Sakip, and Alham Fikri Aji. 2025. Data Laundering: Artificially Boosting Benchmark Results through Knowledge Distillation. In Proceedings of the 63rd Annual Meeting of the Association for Computational Linguistics (Volume 1: Long Papers), pages 8332–8345, Vienna, Austria. Association for Computational Linguistics.

---

### Meta-Review · Area_Chair_GErV · 2026-01-06

**Summary:**

The paper studies whether knowledge distillation (KD) meaningfully transfers test-set leakage from a contaminated teacher to a clean student (“data laundering”). Across eight classification benchmarks, it finds laundering effects are generally small, often insignificant, and much weaker than direct contamination. A key claim is that laundering appears mainly when train–test distribution gaps are large, and KD can often act as a decontamination bottleneck. Main disagreement among reviewers was about scope and generality: whether results based on BERT-style classification meaningfully speak to modern LLMs and generative settings, and whether the conclusions were framed too strongly given small effect sizes.

**Reviewer Concerns:**

1. Model scope (BERT-only, relevance to LLMs): Largely addressed. Authors added experiments with Llama-3.2-1B and Qwen3-0.6B as teachers, showing similar suppression of laundering. Still limited to classification, but this was explicitly re-scoped and clarified.
2. Task scope (classification vs. generative): Partially addressed. No new generative experiments, but authors narrowed claims, updated title, and clearly framed generative settings as future work. Reasonable but not a full resolution.
3. Contamination protocol realism (full test injection, replace vs add): Partially addressed. Authors justify the protocol as a worst-case stress test and explain why partial/fuzzy contamination was avoided. This is a defensible argument, but still an assumption rather than empirical validation.
4. Small effect sizes / over-strong narrative: Mostly addressed. Added confidence intervals, clarified that effects are rare and benchmark-specific, and softened claims in several places.
5. Distillation objectives / actionable guidance: Addressed. Additional appendix analysis and explicit discussion of privacy–utility tradeoffs across KD strategies were added.

**Reviewer Scores:**

Reviewer 7UAR (4 > 5):  Main concerns about scope and framing are partially resolved; still some unease about generalization.
Reviewer Uoo6 (6 > 6):  Already positive; added LLM experiments and benchmark guidance directly address their main asks.
Reviewer Vwqr (6 > 6): Concerns about scope and data-free/generative KD are acknowledged and scoped as limitations; unlikely to downgrade.
Reviewer TbMs (4 > 6): Explicitly updated score after rebuttal; concerns resolved.

---

### Decision · Program_Chairs · 2026-01-26

Accept (Poster)